# Unleashing the Potential of Temperature Scaling for Multi-Label Logit Distillation

## Abstract

This paper undertakes meticulous scrutiny of the *pure logit-based* distillation under *multi-label learning* through the lens of *activation function*. We begin with empirically clarifying a recently discovered perspective (Yang et al., 2023b;a) that vanilla *sigmoid* per se is more suitable than *tempered softmax* in multi-label distillation, is not entirely correct. After that, we reveal that both the *sigmoid* and *tempered softmax* have an intrinsic limitation. In particular, we conclude that ignoring the decisive factor temperature $\tau$ in the *sigmoid* is the essential reason for its unsatisfactory results. With this regard, we propose unleashing the potential of temperature scaling in the multi-label distillation and present Tempered Logit Distillation (TLD), an embarrassingly simple yet astonishingly performant approach. Specifically, we modify the *sigmoid* with the temperature scaling mechanism, deriving a new activation function, dubbed as *tempered sigmoid*. With theoretical and visual analysis, intriguingly, we identify that *tempered sigmoid* with $\tau$ *smaller than* 1 provides an effect of hard mining by governing the magnitude of penalties according to the sample difficulty, which is shown as the key property to its success. Our work is accompanied by comprehensive experiments on COCO, PASCAL-VOC, and NUS-WIDE over several architectures across three multi-label learning scenarios: image classification, object detection, and instance segmentation. Distillation results evidence that TLD consistently harvests remarkable performance and surpasses the prior counterparts, demonstrating its superiority and versatility.

## 1 Introduction

Knowledge distillation (KD), which aims to impart the informative knowledge from a cumbersome teacher ($\mathcal{T}$) model to a lightweight student model ($\mathcal{S}$), is an effective model compression technique (Hinton et al., 2015). In general, KD is executed via minimizing the difference (e.g., Kullback-Leibler ($\mathcal{L}_{KL}$) divergence) between classification responses (i.e., logit) of the teacher and student. A crucial step of such KD is *how to activate the logits*, as depicted in Figure 1. In the common practice, the *tempered softmax* is used to activate the logit, in which a temperature scaling factor $\tau$ is introduced to modulate the logit smoothness. The temperature $\tau$ drastically affects the final KD performance (Zhao et al., 2022; Xu et al., 2020; Li et al., 2022b; Kim et al., 2021; Kobayashi,

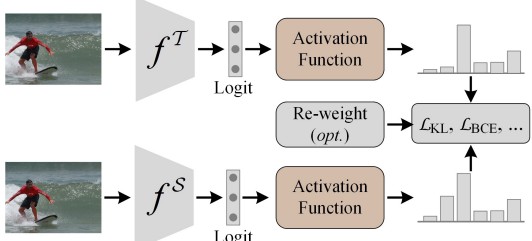

Figure 1: The general distilling pipeline for logit KD. Through unleashing the potential of temperature scaling, this paper introduces a better Activation Function, which is recognized as the foundation for improving multi-label logit distillation.

2022; Chandrasegaran et al., 2022; Li et al., 2023b;a; Sun et al., 2024; Zheng & Yang, 2024) and the sweet spots of $\tau$ is *greater than* 1 ($\tau > 1$), both empirically and theoretically. Over the past few years, KD has found success in plenty of single-label learning tasks (Huang et al., 2022; Yang et al., 2022a; Ding et al., 2023; Miles et al., 2023; Zheng & Yang, 2024).

Recently, applying KD to the more challenging yet realistic multi-label setting has aroused wide attention (Yang et al., 2023b;a; Zhang et al., 2023). Unfortunately, the concurrent publications L2D

(Yang et al., 2023b) in the classification (*Cls.*) and BCKD (Yang et al., 2023a) in the detection (*Det.*) discovered that directly migrating the classical KD, i.e., activating logits with *tempered softmax*, to multi-label tasks results in trivial improvements. Specifically, L2D stated that the predictive probabilities may not equal one in multi-label classification, thus causing *tempered softmax* to be not suitable. BCKD, *based on an overstrict presumption*[1], argued that the *tempered softmax* leads to loss vanishing, curtailing the KD performance eventually. To combat this problem, they activate the logit with the vanilla *sigmoid* and minimize the difference with Binary-Cross Entropy ($\mathcal{L}_{\mathrm{BCE}}$) loss.

Nevertheless, both the exploration in L2D (Yang et al., 2023b) and BCKD (Yang et al., 2023a) lacks a convincing theoretical understanding and comprehensive experimental evidence, therefore leaving how to activate logit in multi-label KD remains an enigma, including: ❶ Whether the vanilla *sigmoid* is truly superior to *tempered softmax* in practice? ❷ What are the limitations of them? ❸ If there is a simple way that could simultaneously make logit KD excel in multi-label learning and provide a theoretically sound justification? In this work, by presenting formal answers to these under-explored questions, we attempt to make a thorough investigation and contribute to the multi-label logit KD.

We start by answering question ❶, and our primary concern is to ascertain whether the decent performance reported in L2D (Yang et al., 2023b) and BCKD (Yang et al., 2023a) can be attributed to *sigmoid*. To this end, we perform pure logit KD on classification and detection tasks. Table 1 summarizes the KD results. Surprisingly, when removing the confounding factors (advanced distance function or extra re-weight strategy[2]), we empirically observe that the performance gain of *sigmoid* substantially disappears and the *tempered sofmax* obtains on par or even better KD results, especially in the classification task, which clearly negates the advantages of vanilla *sigmoid*. As a short takeaway, it seems that the vanilla *sigmoid* itself is not better than *tempered softmax* in multi-label logit KD. Additional experiments in later sections further confirm our findings.

As to the question ❷, we disclose that both the *tempered softmax* and vanilla *sigmoid* have an intrinsic pitfall in multi-label KD. For *tempered softmax*, we expose that it is leveraged to capture and transfer the *inter-*

Table 1: Motivation experiments on the COCO dataset for comparing the activation functions ($\sigma$) in the multi-label distillation. Related results for *Cls.* and *Det.* are respectively collected from L2D (Yang et al., 2023b) and BCKD (Yang et al., 2023a), if reported. The three types of distance function ($\mathcal{D}$) are the most adopted in logit KD. For sample re-weighting (M), we choose the strategy used in BCKD, and ✗ means the M *can not* be applied. * indicates the vanilla distance metric for each activation function. $\Delta$ represents the performance gains over the undistilled student. The teacher-student are ResNet101-ResNet34 and GFocal based ResNet101-ResNet18 for *Cls.* and *Det.*, respectively.

| $\sigma$ | $\mathcal{D}$ | M | *Cls.* | | *Det.* | |
|---|---|---|---|---|---|---|
| | | | mAP | $\Delta$ | mAP | $\Delta$ |
| Teacher | - | - | 73.62 | - | 44.9 | - |
| Student | - | - | 70.31 | - | 35.8 | - |
| *tempered softmax* | $\mathcal{L}_{\mathrm{KL}}{}^{*}$ | - | 71.88 | 1.57 | 36.4 | 0.6 |
| w/ $\tau > 1$ | $\mathcal{L}_{\mathrm{DIST}}$ | ✗ | 72.03 | 1.72 | 36.0 | 0.2 |
| | $\mathcal{L}_{\mathrm{BCE}}{}^{*}$ | - | 70.68 | 0.37 | 36.5 | 0.7 |
| *sigmoid* | $\mathcal{L}_{\mathrm{BCE}}{}^{*}$ | ✓ | 73.41 | 3.10 | 37.2 | 1.4 |
| | $\mathcal{L}_{\mathrm{DIST}}$ | ✗ | 73.43 | 3.13 | 37.2 | 1.4 |
| *tempered sigmoid* | $\mathcal{L}_{\mathrm{KL}}{}^{*}$ | - | 73.48 | 3.17 | 37.3 | 1.5 |
| w/ $\tau < 1$ (Ours) | $\mathcal{L}_{\mathrm{DIST}}$ | ✗ | **74.26** | **3.95** | **37.5** | **1.7** |

*class* relations, which is the primary attribution of such logit KD (Zhao et al., 2022; Chandrasegaran et al., 2022; Li et al., 2022b; Zheng & Yang, 2024). With this perspective, *tempered softmax* is naturally sub-optimal for logit distillation in multi-label learning, in which each class is optimized *individually*. A similar conclusion can also be found in (Li et al., 2022a; Zhang et al., 2023). Intuitively, as claimed in L2D (Yang et al., 2023b) and BCKD (Yang et al., 2023a), the one-versus-all (OVA) approach, i.e., *sigmoid*, in multi-label learning can alleviate this problem. However, our empirical results imply that the vanilla *sigmoid* per se does not have the capacity as expected. Something must be largely overlooked. In this paper, we posit that ignoring $\tau$ (i.e., $\tau$ is fixed as 1) is the essential reason for undermining the potential power of *sigmoid*. This argument is partially underpinned by the constant superior results established in single-label learning tasks (Li et al., 2022b; 2023b; Sun et al., 2024; Zheng & Yang, 2024) by analyzing the role of $\tau$ in KD. Besides, the exploratory experiments in Table 1 directly attest to our conjecture that the temperature scaled *sigmoid*, termed *tempered sigmoid*, could lead to better results in multi-label KD.

---

[1]BCKD introduced a constant tensor $\eta$ to meet a presumption: $\mathbf{z}^s = \mathbf{z}^t + \eta$. $\mathbf{z}$ is the logit. However, such a presumption *can not* be held for each sample/pixel. Interested readers may refer to their paper for more details.

[2]The re-weight strategy is widely used in the detection task. Hence, to fairly compare *sigmoid* and *tempered softmax*, we also include the KD results without this strategy. BCKD *did not* conduct such an ablative study.

Based on the above analysis, we are strongly encouraged to take the *tempered sigmoid* as a response to the question ❸. Building upon the behavior of multi-label learning(Yang et al., 2023b) and temperature scaling (Li et al., 2022b; 2023b), *tempered sigmoid* brings two desirable properties: 1) activate each class in the logit *individually*, successfully aligning the training protocol of the original tasks; 2) bring back the critical factor $\tau$, endowing us to unleash its effects for multi-label KD. Particularly, the *tempered sigmoid* derives two intriguing findings regarding the selection of $\tau$ in logit KD. First, a lower value ($\tau < 1$) achieves more satisfactory performances with *tempered sigmoid*, which is exactly opposite to the optimal setting ($\tau > 1$) for *tempered softmax*. One is encouraged to think about the reason. The theoretical and empirical answers will be provided later. Besides, we find that L2D and BCKD are the special case of the proposed method but with $\tau = 1$.

Our method also possesses two appealing merits: it is compatible with different distance functions (i.e., $\mathcal{L}_{\mathrm{KL}}$ or $\mathcal{L}_{\mathrm{DIST}}$); it is flexible cooperating with feature-based approaches. Massive empirical results on COCO (Lin et al., 2014), PASCAL-VOC (Everingham et al., 2015), and NUS-WIDE (Chua et al., 2009) evidence that our method achieves state-of-the-art performance on image classification, object detection, and instance segmentation tasks. The embarrassing simplicity and excellent performance of the proposed method may foresee its broad application in multi-label logit KD and warrant future research. To sum up, this paper makes the following contributions:

- We clarify that the vanilla *sigmoid* per se does not bring significant practical benefits than *tempered softmax* in multi-label logit distillation, as claimed in previous publications. The limitation of vanilla *sigmoid* and *tempered softmax* is further discussed.

- In particular, we disclose that ignoring the temperature parameter is the essential bottleneck causing the vanilla *sigmoid* to produce poor results. This motivates us to introduce the *tempered sigmoid*, which is theoretically shown as a hardness-aware activation function with a proper $\tau$ in multi-label distillation.

- In light of the general character of *tempered sigmoid*, our method adapts surprisingly well to three prevalent multi-label learning tasks, and is seamlessly embedded with other distillation approaches to further enhance KD.

- Despite its simplicity, the proposed method consistently delivers notable performance gains in multi-label distillation and beats its prior counterparts by a large margin.

## 2 RELATED WORK

**Multi-Label Learning.** Multi-label learning, in which each sample is generally associated with multiple class labels, is more applicable for real-world applications (Yang et al., 2023b). Thanks to the recent advance of backbones (e.g., ResNet (He et al., 2016) and Swin (Liu et al., 2021)), multi-label learning supports several visual recognition tasks, including image classification (Chen et al., 2019b), object detection (Lin et al., 2017b), and instance segmentation (Wang et al., 2020).

**Knowledge Distillation.** Knowledge distillation (KD), which aims to learn a compact yet powerful model by inheriting knowledge from a high-capacity one, is popularized in single-label learning (Hinton et al., 2015). Several papers have attempted to investigate KD in multi-label learning tasks. L2D (Yang et al., 2023b) studied this problem in the classification task and introduced a one-versus-all distillation manner. In object detection, BCKD (Yang et al., 2023a) further formulated the logit maps as multiple binary-classification maps with *sigmoid* and then performed distillation. However, these methods left some critical questions unanswered and did not explore this problem in general.

**Temperature Scaling in KD.** In temperature scaling, a parameter $\tau$ is involved to adjust the slope of predictive distributions. Its pivotal role and practical benefits in logit KD are justified with sufficient evidence (Xu et al., 2020; Kim et al., 2021; Chandrasegaran et al., 2022; Li et al., 2022b; Kobayashi, 2022; Li et al., 2023b; Zheng & Yang, 2024; Sun et al., 2024; Wang et al., 2024). For example, ATS (Li et al., 2022b) teaches better students by applying different $\tau$ to the different classes. CTKD (Li et al., 2023b) adjusted the $\tau$ through an easy-to-hard curriculum learning strategy. TTM (Zheng & Yang, 2024) dropped the temperature $\tau$ on the student side, further boosting the standard logit KD. LSKD (Sun et al., 2024) assigned distinct $\tau$ between the teacher and student and across samples. However, **all** these methods are based on *tempered softmax* and devoted to single-label classification KD. This paper, with distinct motivation, explores the specific properties of $\tau$ in multi-label distilla-

tion and introduces *tempered sigmoid*. In addition, using the $\tau$ as a proxy, we recognize and verify some intriguing phenomena among different logit KD under different learning tasks.

**Hard Samples Mining in KD.** Mining and prioritizing hard/important samples is a long-standing issue in knowledge distillation. For example, (Hu et al., 2024; Li et al., 2022a; Yang et al., 2023a) find that the samples exhibiting larger discrepancy between the teacher and student is hard for distillation and paying more learning effort on them favors the student. This issue is more critical for the detection KD, where the training samples have an extreme imbalance between positive and negative pixels. To alleviate this, numerous methods propose to distill the "knowledge-dense" locations by various sophisticatedly-designed strategies (Wang et al., 2019; Dai et al., 2021; Huang et al., 2023; Li et al., 2022a; Yang et al., 2022b; Zheng et al., 2022). For instance, (Wang et al., 2019) selected anchors overlapping with the ground-truth object anchors as distillation regions. (Huang et al., 2023) leveraged multiple receptive tokens to perceive each pixel via attention calculation, generating a pixel-wise distillation mask. (Dai et al., 2021; Li et al., 2022a) distills student detectors based on valuable locations selected by the predictive discrepancy between the teacher and student. In this paper, we revealed that the simple temperature scaling operation is capable of providing the effect of hard mining and consistently outperforms the previous publications.

## 3 METHODOLOGY

We elaborate our method under the classification task for simplicity, and extending it to dense prediction tasks is straightforward. Let $\mathbf{z} = [z_1, z_2, ..., z_c] \in \mathbb{R}^C$ represent the logit, where $z_i$ is the logit output of the $i$-th class and $C$ is the number of classes. With an activation function $\sigma$, $\mathbf{z}$ is converted to the predictive probability $\mathbf{p} \in \mathbb{R}^C$. We discriminate the notations with superscripts $s$ and $t$ for the student ($\mathcal{S}$) and teacher ($\mathcal{T}$).

### 3.1 PRELIMINARIES

**Tempered Softmax.** Following the classical KD (Hinton et al., 2015) in single-label learning, the logit is activated by *softmax* with a temperature factor $\tau$:

$$\mathbf{p}_{i,\tau} = \text{softmax}(z_i, \tau) = \frac{e^{z_i/\tau}}{\sum_{c=1}^{C} e^{z_c/\tau}} \tag{1}$$

and the KD is implemented by minimizing the Kullback-Leibler ($\mathcal{L}_{\text{KL}}$) divergence :

$$\mathcal{L}_{\text{KL}}(\mathbf{p}_\tau^s, \mathbf{p}_\tau^t) = -\tau^2 \sum_{i=1}^{C} \mathbf{p}_{i,\tau}^t \log \mathbf{p}_{i,\tau}^s \tag{2}$$

where $\tau$ is involved to regulate the logit smoothness, and generally, $\boldsymbol{\tau > 1}$ (e.g., $\tau = 10$ in LD (Zheng et al., 2022)). Note that, we omit the denominator term in $\mathcal{L}_{\text{KL}}$ as it does not contribute to the gradient. We refer to this method as KL.

**Sigmoid.** Lately, L2D (Yang et al., 2023b) argued that the above KD scheme in single-label learning can not be directly applied in the multi-label scenario, and advocated that the *sigmoid* activated logit is more suitable for performing distillation. A similar conclusion can be also drawn in BCKD (Yang et al., 2023a) with the detection task. Formally, *sigmoid* activates the logit as:

$$\mathbf{p}_i = \text{sigmoid}(z_i) = \frac{1}{1 + e^{-z_i}} \tag{3}$$

and KD loss is calculated by Binary Cross Entropy ($\mathcal{L}_{\text{BCE}}$), and we refer to this method as BCE.

$$\mathcal{L}_{\text{BCE}}(\mathbf{p}^s, \mathbf{p}^t) = -\sum_{i=1}^{C} (\mathbf{p}_i^t \log \mathbf{p}_i^s + (1 - \mathbf{p}_i^t) \log(1 - \mathbf{p}_i^s)) \tag{4}$$

However, as discussed before, the *tempered softmax* and vanilla *sigmoid* both have an intrinsic problem regarding performing KD in multi-label learning tasks. Our experimental results also prove

that they actually distill similar inferior students in practice, as shown in Table 1. In this paper, we put forth an alternative activation manner through a simple yet effective temperature scaling operation that can attain better performance and provide convincing justification.

## 3.2 TEMPERED LOGIT DISTILLATION (TLD)

**Loss Formulation.** We reduce the multi-label classification score to a series of *independent* binary classification tasks. Formally, we obtain the positive response of the $i$-th class with *tempered sigmoid* as:

$$\mathbf{p}_{i,\tau} = \text{sigmoid}(z_i, \tau) = \frac{1}{1 + e^{-z_i/\tau}} \tag{5}$$

where $\tau$ plays the same role as in Eq. (1). Under the OVA strategy, we derive the negative response for the $i$-th class as $1 - \mathbf{p}_{i,\tau} = \frac{1}{1+e^{z_i/\tau}}$. By concatenating the positive and negative class response, we have the binary predicted probabilities $\tilde{\mathbf{p}}_\tau = [[\mathbf{p}_{1,\tau}, 1 - \mathbf{p}_{1,\tau}], ..., [\mathbf{p}_{c,\tau}, 1 - \mathbf{p}_{c,\tau}]] \in \mathbb{R}^{C \times 2}$. Then we distill the *per-class independently* with a binary version of $\mathcal{L}_{\text{KL}}$ and the KD loss is as:

$$
\begin{aligned}
\mathcal{L}_{\text{TLD}} = \mathcal{L}_{\text{KL}}(\tilde{\mathbf{p}}_\tau^s, \tilde{\mathbf{p}}_\tau^t) &= \sum_{i=1}^{C} \mathcal{L}_{\text{KL}}(\tilde{\mathbf{p}}_{i,\tau}^s, \tilde{\mathbf{p}}_{i,\tau}^t) \\
&= -\tau^2 \sum_{i=1}^{C} (\mathbf{p}_{i,\tau}^t \log \mathbf{p}_{i,\tau}^s + (1 - \mathbf{p}_{i,\tau}^t) \log(1 - \mathbf{p}_{i,\tau}^s))
\end{aligned}
\tag{6}
$$

Note that, we choose the vanilla $\mathcal{L}_{\text{KL}}$ as the distance function in TLD to make fair comparisons with KL and BCE. However, TLD works well with more advanced loss functions, e.g., $\mathcal{L}_{\text{DIST}}$ in (Huang et al., 2022). Furthermore, comparing Eq. (4) with Eq. (6), we can derive that L2D (Yang et al., 2023b) and BCKD (Yang et al., 2023a) is equivalent to our method by setting $\tau = 1$.

**Overall Loss.** The overall training objective of optimizing the student is formulated as:

$$\mathcal{L}_{\text{overall}} = \mathcal{L}_{\text{task}} + \lambda \mathcal{L}_{\text{TLD}} \tag{7}$$

where $\mathcal{L}_{\text{task}}$ is the original task loss for training the students and $\lambda$ is the factor for balancing the losses. In practical scenarios, any *pure feature-based* methods in the corresponding task can be easily fused to gain extra improvements. Distillation results will be included for this consideration.

**Theoretical Analysis.** Here, we theoretically show that TLD pays more attention to hard samples during KD when the $\tau$ is properly set, i.e., $\tau < 1$. Since our TLD distill per-class individually, we derive (more details are in Appendix C.1) the gradient of $\mathcal{L}_{\text{TLD}}$ with respect to $z_i^s$ as follows and:

$$\frac{\partial \mathcal{L}_{\text{TLD}}}{\partial z_i^s} = \tau(\tilde{\mathbf{p}}_{i,\tau}^s - \tilde{\mathbf{p}}_{i,\tau}^t) \tag{8}$$

This formula suggests that the gradient is proportional to the discrepancy between $\tilde{\mathbf{p}}_{i,\tau}^s$ and $\tilde{\mathbf{p}}_{i,\tau}^t$, with $\tau$ controlling the penalty strength of each sample. Now, let us consider two cases.

(i) When $\tau < 1$ and $\tau \to 0$, the $\tilde{\mathbf{p}}_{i,\tau}^t$ starts resembling to the ground-truth $[1, 0]$. Since the student could make correct (i.e., ground-truth) predictions $\tilde{\mathbf{p}}_{i,\tau}^s$ for easy samples, therefore, the loss magnitude that comes from the hard samples is intensified with $\tau < 1$. In other words, our TLD under $\tau < 1$ regulates the students to take more effort into emulating the hard areas, benefiting the KD performance eventually. Note that an extremely small $\tau$ will cause the loss only to concentrate on very limited samples, thus degrading the performance as well.

(ii) Conversely, when $\tau \geq 1$ and $\tau \to \infty$, the $\tilde{\mathbf{p}}_{i,\tau}^t$ is approaching to a uniform one, i.e., $[0.5, 0.5]$. In this case, the loss tends to distribute over the whole logit space, accordingly attenuating the learning effort put in hard samples. Besides, uniform targets render the identity information erasure, which also negatively affects the student's learning.

Table 2: Multi-Label image classification KD performance on COCO. $^{\dagger}$ indicates that we replace the vanilla $\mathcal{L}_{\mathrm{KL}}$ with more advanced $\mathcal{L}_{\mathrm{DIST}}$ (Huang et al., 2022) in Eq. (6). The related results reference the L2D (Yang et al., 2023b).

| Method | ResNet101-ResNet34 | | | SwinTiny-MobileNetV2 | | | RepVGGA2-RepVGGA0 | | | WRN101-WRN50 | | |
|---|---|---|---|---|---|---|---|---|---|---|---|---|
| | mAP | OF1 | CF1 | mAP | OF1 | CF1 | mAP | OF1 | CF1 | mAP | OF1 | CF1 |
| Teacher | 73.62 | 73.89 | 68.61 | 79.43 | 78.77 | 75.07 | 72.71 | 74.11 | 68.63 | 74.70 | 75.76 | 70.73 |
| Student | 70.31 | 72.49 | 66.82 | 71.85 | 73.59 | 68.26 | 70.02 | 72.49 | 66.77 | 74.45 | 75.43 | 70.61 |
| RKD | 70.13 | 72.44 | 66.78 | 71.74 | 73.68 | 68.37 | 70.13 | 72.39 | 66.73 | 74.70 | 75.71 | 70.84 |
| PKT | 70.43 | 72.64 | 66.68 | 71.84 | 73.76 | 68.37 | 70.11 | 72.47 | 66.80 | 74.54 | 75.47 | 70.58 |
| Review KD | 70.39 | 72.62 | 66.76 | 71.73 | 73.71 | 68.33 | 70.00 | 72.35 | 66.82 | 74.03 | 75.29 | 70.36 |
| MSE | 70.54 | 72.75 | 66.85 | 71.80 | 73.74 | 68.38 | 70.26 | 72.54 | 66.99 | 74.53 | 75.60 | 70.71 |
| PS | 70.86 | 72.66 | 67.12 | 72.42 | 74.14 | 68.94 | 70.65 | 72.89 | 67.60 | 75.12 | 76.05 | 71.63 |
| BCE | 70.68 | 72.69 | 67.19 | 72.35 | 74.10 | 68.91 | 70.74 | 72.81 | 67.46 | 74.92 | 75.75 | 71.21 |
| L2D | 72.02 | 73.63 | 68.27 | 73.45 | 75.19 | 69.92 | 72.17 | 74.00 | 68.85 | 74.82 | 75.79 | 71.25 |
| TLD | 73.48 | 74.84 | 69.37 | 73.93 | 75.21 | 70.45 | 72.71 | 74.34 | 69.20 | 77.29 | 77.65 | 72.97 |
| TLD$^{\dagger}$ | 74.26 | 75.32 | 70.06 | 74.08 | 75.47 | 70.69 | 73.01 | 74.48 | 69.31 | 77.40 | 77.71 | 73.16 |
| BCE + L2D | 72.87 | 74.45 | 69.43 | 74.21 | 75.72 | 70.87 | 72.81 | 74.59 | 69.49 | 76.61 | 77.08 | 72.79 |
| TLD + L2D | 74.01 | 75.14 | 70.23 | 74.75 | 75.94 | 70.95 | 73.52 | 74.99 | 69.69 | 77.59 | 77.91 | 73.32 |

Table 3: Multi-Label image classification KD performance on PASCAL-VOC. The related results reference the L2D (Yang et al., 2023b).

| Method | ResNet50-ResNet18 | | | SwinSmall-SwinTiny | | |
|---|---|---|---|---|---|---|
| | mAP | OF1 | CF1 | mAP | OF1 | CF1 |
| Teacher | 86.73 | 84.92 | 81.21 | 92.75 | 91.05 | 88.82 |
| Student | 84.01 | 83.60 | 79.42 | 91.31 | 89.98 | 88.00 |
| RKD | 84.48 | 83.54 | 79.83 | 91.52 | 90.44 | 88.51 |
| PKT | 84.12 | 83.10 | 79.31 | 91.28 | 90.17 | 88.03 |
| Review KD | 83.71 | 83.01 | 79.25 | 91.45 | 90.17 | 88.06 |
| MSE | 84.23 | 83.16 | 79.29 | 91.06 | 89.99 | 87.66 |
| PS | 84.44 | 83.78 | 79.95 | 91.21 | 90.25 | 88.12 |
| BCE | 84.48 | 84.07 | 80.29 | 91.43 | 90.25 | 88.12 |
| L2D | 85.42 | 85.08 | 81.46 | 91.65 | 90.84 | 88.96 |
| TLD | 86.32 | 85.56 | 82.03 | 92.59 | 91.34 | 89.44 |
| BCE + L2D | 85.71 | 85.70 | 82.11 | 91.92 | 91.34 | 89.58 |
| TLD + L2D | 86.70 | 85.72 | 82.35 | 92.80 | 91.47 | 89.81 |

Table 4: Multi-Label image classification KD performance on NUS-WIDE. The related results reference the L2D (Yang et al., 2023b).

| Method | ResNet101-ResNet34 | | | SwinTiny-MobileNetV2 | | |
|---|---|---|---|---|---|---|
| | mAP | OF1 | CF1 | mAP | OF1 | CF1 |
| Teacher | 55.32 | 75.56 | 61.13 | 59.73 | 77.30 | 65.44 |
| Student | 53.41 | 75.10 | 60.08 | 54.49 | 75.72 | 61.74 |
| RKD | 53.62 | 75.20 | 59.91 | 54.76 | 75.69 | 61.74 |
| PKT | 53.55 | 75.08 | 60.35 | 54.59 | 75.69 | 61.74 |
| Review KD | 53.52 | 75.23 | 60.44 | 54.85 | 75.84 | 61.75 |
| MSE | 53.52 | 75.13 | 59.94 | 54.86 | 75.80 | 61.69 |
| PS | 54.14 | 75.43 | 60.79 | 55.18 | 75.91 | 62.35 |
| BCE | 54.44 | 75.36 | 60.73 | 55.36 | 76.00 | 62.52 |
| L2D | 53.89 | 75.19 | 61.07 | 56.24 | 76.50 | 63.15 |
| TLD | 55.24 | 75.68 | 60.60 | 56.91 | 76.76 | 63.43 |
| BCE + L2D | 55.31 | 76.17 | 62.79 | 56.91 | 76.92 | 63.89 |
| TLD + L2D | 56.83 | 76.64 | 63.52 | 57.86 | 77.30 | 64.44 |

## 4 EXPERIMENTS

Our experiments are conducted on three standard benchmark datasets, i.e., COCO (Lin et al., 2014), PASCAL-VOC (Everingham et al., 2015), and NUS-WIDE (Chua et al., 2009), across three multi-label learning tasks, including image classification, object detection, and instance segmentation.

Due to the page limitation, we attach all the experimental settings, e.g., datasets, evaluation metrics, networks, training details, and the references of the related methods, in Appendix A.

### 4.1 IMAGE CLASSIFICATION

**Comparison with State-of-the-Arts.** We first validate TLD on the classification task. Table 2, Table 3, and Table 4 list the results. On average, the distilled student achieves outstanding 2~3% absolute gains, which even outstrip the teacher in some cases, and extra improvements can be achieved with an advanced loss function in (Huang et al., 2022). Furthermore, by only distilling with the logit, TLD delivers competitive performance and surpasses prior sophisticated approaches with considerable margins, strongly demonstrating its superiority and effectiveness. Moreover, cooperating with the feature-based L2D, our TLD can be further enhanced and establishes a new state-of-the-art in the multi-label classification KD community. Following L2D, Figure 2 visualizes the class activation maps. It is observed that our TLD precisely locates the discriminative regions, especially for the hard instances, eventually resulting in higher performance. In Appendix B.2, we provide more KD results (Tables B.2, B.3), class-wise performance (Figure B.1), and visualizations (Figure B.2).

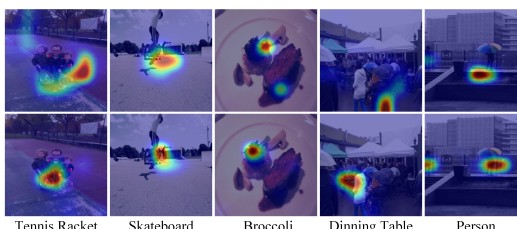

Tennis Racket    Skateboard    Broccoli    Dinning Table    Person

Figure 2: Visualizations of the class activation maps generated from L2D (top) and TLD (bottom). The texts bellow the images are the query categories. The models are the distilled ResNet34.

Table 5: Relation between the student and teacher on multi-label KD. We set ResNet18 as the student and vary teachers.

| Teacher | | Student | | |
|---|---|---|---|---|
| Backbone | mAP | mAP | OF1 | CF1 |
| ResNet18 | 67.81 | 69.28 | 71.87 | 66.19 |
| ResNet34 | 70.40 | 69.68 | 72.20 | 66.65 |
| ResNet50 | 73.36 | 70.58 | 72.77 | 67.24 |
| ResNet101 | 73.62 | 71.19 | 73.21 | 67.44 |
| ResNet152 | **74.97** | **71.23** | **73.26** | **67.61** |

Table 6: Comparison with $\sigma$ on object detection. The results are reported on COCO.

| Method | mAP | $AP_{50}$ | $AP_{75}$ | $AP_S$ | $AP_M$ | $AP_L$ |
|---|---|---|---|---|---|---|
| $\mathcal{T}$: GF-R101 | 44.9 | 63.1 | 49.0 | 28.0 | 49.1 | 57.2 |
| $\mathcal{S}$: GF-R18 | 35.8 | 53.1 | 38.2 | 18.9 | 38.9 | 47.9 |
| KL | 36.4 | 53.9 | 39.0 | 19.6 | 39.8 | 48.3 |
| BCE | 36.5 | 54.0 | 39.0 | 19.7 | 40.0 | 48.2 |
| TLD | **37.3** | **55.4** | **40.1** | **20.4** | **40.9** | **48.9** |
| $\mathcal{S}$: GF-R34 | 38.9 | 56.6 | 42.2 | 21.5 | 42.8 | 51.4 |
| KL | 39.8 | 57.8 | 42.1 | 22.1 | 44.2 | 52.4 |
| BCE | 39.6 | 57.5 | 42.8 | 22.4 | 43.8 | 51.8 |
| TLD | **40.4** | **58.9** | **43.3** | **23.4** | **44.7** | **52.7** |
| $\mathcal{S}$: GF-R50 | 40.2 | 58.4 | 43.3 | 23.3 | 44.0 | 52.2 |
| KL | 41.0 | 59.4 | 44.3 | 23.6 | 45.0 | 53.0 |
| BCE | 40.7 | 59.2 | 43.9 | 23.4 | 44.7 | 53.2 |
| TLD | **41.9** | **60.8** | **45.3** | **24.9** | **46.0** | **54.4** |

Table 7: Comparison with logit KD on object detection. The results are reported on COCO.

| Method | mAP | $AP_{50}$ | $AP_{75}$ | $AP_S$ | $AP_M$ | $AP_L$ |
|---|---|---|---|---|---|---|
| $\mathcal{T}$: AT-R50 | 39.4 | 57.6 | 42.8 | 23.6 | 42.9 | 50.3 |
| $\mathcal{S}$: AT-R18 | 34.8 | 53.1 | 37.1 | 19.3 | 37.9 | 45.6 |
| CWD | 35.7 | 53.1 | 38.6 | 19.5 | 38.8 | 47.3 |
| RM | 35.4 | 53.3 | 37.9 | 19.3 | 38.6 | 46.1 |
| DIST | 35.7 | 53.3 | 38.5 | 18.9 | 39.0 | 46.7 |
| TLD | **36.7** | **54.5** | **39.5** | **20.4** | **39.6** | **47.8** |
| $\mathcal{T}$: GF-R50 | 40.2 | 58.4 | 43.3 | 23.3 | 44.0 | 52.2 |
| $\mathcal{S}$: GF-R18 | 35.8 | 53.1 | 38.2 | 18.9 | 38.9 | 47.9 |
| CWD | 36.5 | 53.7 | 39.2 | 18.4 | 40.1 | 48.4 |
| RM | 36.4 | 53.8 | 38.7 | 19.8 | 39.3 | 48.0 |
| DIST | 36.4 | 53.8 | 39.1 | 18.8 | 40.1 | 48.4 |
| TLD | **37.1** | **54.9** | **39.8** | **19.9** | **40.9** | **48.9** |

**Better Teacher, Better Student.** In single-label logit KD, (Mirzadeh et al., 2020; Huang et al., 2022) found a counter-intuitive phenomenon that a stronger teacher may harm the performance of students, i.e., *better teacher, worst student*. We study this issue in multi-label logit KD. As shown in Table 5, there is a clear positive correlation between the mAP of the student and teachers, and a better teacher tends to teach a better student in multi-label KD. Our method seems to enjoy the large discrepancy between the teacher and student, i.e., *better teacher, better student*.

## 4.2 Object Detection

**Comparison with $\sigma$.** As we focus on the activation function $\sigma$ regarding logit KD, we first compare the performance of existing methods to ours, and the corresponding results are listed in Table 6. It is observed that TLD delivers notable performance enhancements, manifesting the effectiveness of our method. For instance, distilled with GFocal-ResNet101, the ResNet18, ResNet34, and ResNet50 based students obtain +1.5%, +1.5%, and +1.7% mAP gains, respectively.

More importantly, TLD consistently outperforms KL and BCE with a clear margin, showing its superiority. In Figure 3, we plot the training stimulus. We can see that the loss distribution of KL (*tempered softmax*) is diverse, almost covering the whole logit map, which hinders the students from distilling knowledge from informative areas. BCE could filter out some background but still fails to discriminate the most valuable pixels. Our TLD (*tempered sigmoid*), however, clearly lays a tight emphasis on the potential seman-

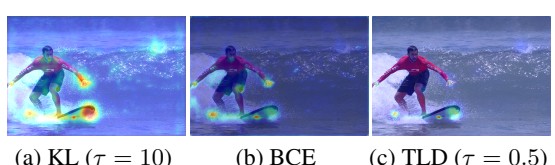

(a) KL ($\tau = 10$)    (b) BCE    (c) TLD ($\tau = 0.5$)

Figure 3: KD loss distribution in the logit space, and the value is normalized to $[0, 1]$. The teacher-student pair is GFocal with ResNet101 and ResNet18.

tic locations and removes the futile background regions, which directly translates to the elevation in performance. In Appendix B.3, Table B.4 summarizes more KD results with various detectors.

Table 8: Comparison with feature KD on object detection. The results are reported on COCO.

| Method | mAP | $AP_{50}$ | $AP_{75}$ | $AP_S$ | $AP_M$ | $AP_L$ |
|---|---|---|---|---|---|---|
| $\mathcal{T}$: GF-R101 | 44.9 | 63.1 | 49.0 | 28.0 | 49.1 | 57.2 |
| $\mathcal{S}$: GF-R50 | 40.2 | 58.4 | 43.3 | 23.3 | 44.0 | 52.2 |
| FitNet | 40.7 | 58.6 | 44.0 | 23.7 | 44.4 | 53.2 |
| GT Box | 40.7 | 58.6 | 44.2 | 23.1 | 44.5 | 53.5 |
| FGFI | 41.1 | 58.8 | 44.8 | 23.3 | 45.4 | 53.1 |
| MasKD | 40.4 | 58.4 | 43.6 | 23.5 | 44.0 | 52.9 |
| DeFeat | 40.8 | 58.6 | 44.3 | 24.3 | 44.6 | 53.7 |
| GID | 41.5 | 59.6 | 45.2 | 24.3 | 45.7 | 53.6 |
| FGD | 41.3 | 58.8 | 44.8 | 24.5 | 45.6 | 53.0 |
| MGD | 42.1 | 60.3 | 45.8 | 24.4 | 46.2 | 54.7 |
| SSKD | 42.3 | 60.2 | 45.9 | 24.5 | 46.7 | 55.6 |
| PKD | 42.5 | 60.9 | 46.0 | 24.2 | 46.7 | 55.9 |
| TLD | 41.9 | 60.8 | 45.3 | 24.9 | 46.0 | 54.4 |
| TLD + FitNet | 42.4 | 60.8 | 45.8 | 24.5 | 46.5 | 55.0 |
| TLD + MGD | 42.3 | 60.8 | 45.8 | 24.4 | 46.4 | 54.8 |
| TLD + PKD | **43.4** | **61.4** | **47.0** | **25.2** | **47.9** | **56.9** |

**Comparison with Other Logit KD.** There are other efforts to promote the logit detection KD, e.g., the channel-wise KD (CWD) (Shu et al., 2021) or rank mimicking (RM) (Li et al., 2022a). Table 7 compares these schemes. The results showcase that our method consistently beats them, indicating TLD is a more effective method for distilling the logit map in object detection.

**Comparison with Feature KD.** As verified in prior logit-based publications (Zheng et al., 2022; Yang et al., 2023a; Huang et al., 2022; Zhao et al., 2022), feature-based methods are naturally superior in distilling detectors. However, as shown in Table 8, our TLD could attain competitive detection enhancements and surpass most of the feature KD methods *devised for distilling detectors*. Additionally, our approach is complementary to the feature KD and can be combined with them to further promote student performance. For example, paired with FitNet (Adriana et al., 2015), *the most plain feature KD for distilling detectors*, we can even exceed the state-of-the-art feature KD methods. Besides, our method accomplishes additional +0.2% and +0.9% mAP improvements over MGD (Yang et al., 2022c) and PKD (Cao et al., 2022), respectively. An interesting finding is that TLD leads to balanced improvements on different instance scales, while the feature KD methods perform better in large objects. The possible reason is that the feature KD methods are typically foreground-oriented, thereby distilling more information for large instances. In contrast, our method is hardness-aware and can excavate the beneficial areas, regardless of the object sizes, accordingly leading to balanced results. The loss distribution on multi-scale in Figure 5 backs up our conjecture.

### 4.3 INSTANCE SEGMENTATION

We further extend TLD to instance segmentation and conduct experiments on SOLOv2 (Wang et al., 2020). As shown in Table 9, our TLD consistently outperforms the prior logit-based competitors and yields comparable results to the strong feature-based methods (i.e., PKD and MGD). Moreover, once again, we beat the state-of-the-art KD methods by introducing FitNet, and MGD realizes additional mAP gains (+0.6%) by incorporating our method.

### 4.4 ABLATION ANALYSIS

**Selection of $\tau$.** As described in prior sections, the temperature $\tau$ is a critical factor in deciding the final KD performance. Here, we show its different impact in KL (*tempered softmax*) and our TLD (*tempered sigmoid*) in multi-label KD. Table 10 lists the results by varying $\tau$. Evidently, there is a trade-off in the selection of $\tau$, and the following observations could be made. (i) For KL, $\tau > 1$ and increasing $\tau$ produces better KD results. However, despite this same conclusion drawn in prior single-label KD works, interestingly, we observe a special case of $\tau < 1$ also delivering a comparable result, and the accuracy saturates after $\tau > 4$. The worst result is obtained with $\tau = 0.25$. (ii) TLD exhibits the exact opposite performance trend with KL. Specifically, it is observed

Table 9: Instance segmentation KD performance on COCO.

| Method | mAP | $AP_{50}$ | $AP_{75}$ | $AP_S$ | $AP_M$ | $AP_L$ |
|---|---|---|---|---|---|---|
| $\mathcal{T}$: SOLOv2-R50 | 34.8 | 54.9 | 36.9 | 13.4 | 37.8 | 53.7 |
| $\mathcal{S}$: SOLOv2-R18 | 30.8 | 49.6 | 32.4 | 10.8 | 32.9 | 49.1 |
| KL | 31.4 | 50.6 | 32.9 | 10.6 | 33.3 | 50.1 |
| BCE | 31.3 | 50.5 | 33.0 | 10.2 | 33.4 | 50.1 |
| CWD | 31.9 | 51.0 | 33.7 | 10.3 | 33.9 | 51.7 |
| DIST | 31.8 | 51.2 | 33.5 | 10.3 | 34.0 | 50.5 |
| PKD | 31.8 | 50.6 | 33.6 | 10.8 | 33.9 | 50.7 |
| MGD | 32.5 | 51.5 | 34.6 | 11.4 | 35.3 | 51.8 |
| TLD | 32.3 | 52.1 | 34.1 | 10.9 | 34.5 | 51.1 |
| TLD + FitNet | 32.8 | 52.1 | 34.7 | 11.2 | 35.1 | 51.8 |
| TLD + MGD | **33.1** | **52.5** | **34.9** | **11.5** | **35.6** | **52.2** |

Table 10: Impact of $\tau$ in KL (*softmax*) and our TLD (*sigmoid*) on COCO.

| $\tau$ | KL | | TLD | |
|---|---|---|---|---|
| | *Cls.* | *Det.* | *Cls.* | *Det.* |
| 0.25 | 70.69 | 36.0 | 73.14 | 36.7 |
| 0.5 | 71.12 | 36.0 | **73.48** | **37.3** |
| 0.75 | 71.38 | 36.4 | 73.26 | 37.2 |
| 2.0 | 72.33 | 36.3 | 71.68 | 36.6 |
| 4.0 | **72.57** | **36.5** | 70.68 | 36.4 |
| 10.0 | 71.88 | 36.4 | 70.12 | 36.3 |

Table 11: Performance of TLD under the self-KD strategy. The detector for *Det.* is GFocal (Li et al., 2020).

| $\mathcal{S}$ | TLD | Cls. | | | Det. | | |
|---|---|---|---|---|---|---|---|
| | | mAP | OF1 | CF1 | mAP | $AP_{50}$ | $AP_{75}$ |
| R18 | | 67.81 | 70.56 | 64.26 | 35.8 | 53.1 | 38.2 |
| | ✓ | 69.28 | 71.87 | 66.19 | 36.2 | 53.6 | 38.7 |
| R34 | | 70.40 | 72.66 | 66.88 | 38.9 | 56.6 | 42.2 |
| | ✓ | 72.94 | 74.49 | 69.13 | 40.0 | 57.8 | 43.2 |
| R50 | | 73.36 | 74.49 | 69.54 | 40.2 | 58.4 | 43.3 |
| | ✓ | 75.74 | 76.48 | 71.87 | 40.8 | 59.2 | 44.4 |

that the mAP drops rapidly when $\tau > 1$ and as $\tau$ increases. TLD distills better students under $\tau < 1$ (even with a very low $\tau$ of 0.25) and keeps relatively robust distillation results on $\tau \in [0.25, 1)$. In Figure 4, we visually back up the above KD results under the dense detection task. (i) When $\tau > 1$, the loss distribution is similar across various $\tau$ for KL and our TLD, therefore leading to similar distillation results. (ii) When $\tau < 1$, the KD loss of our TLD is quickly dominated by the areas, e.g., the hand of *person* and edge for *surfboard*. Those regions are supposed to be informative for distillation (Zheng et al., 2022), consequently guaranteeing superior performance by mimicking them. This observation is consistent with the theoretical analysis before. It is noteworthy that $\tau = 0.25$ produces the most sparse imitation regions with TLD but still gives us an acceptable result and *beats the best of KL*. Such surprising results clearly manifest that TLD with $\tau < 1$ regulating the students to tilt distilling the most conducive samples can achieve better results. In comparison, the loss generated by KL (w/ $\tau < 1$) displays a broad impact and starts assigning large magnitudes of penalties to the futile background, which may shed some light on its inferior performance.

**Self-KD.** In self-KD, the teacher and student share the network architectures. Here, we evaluate our method under this special KD paradigm. As reported in Table 11, our TLD can still bring noticeable

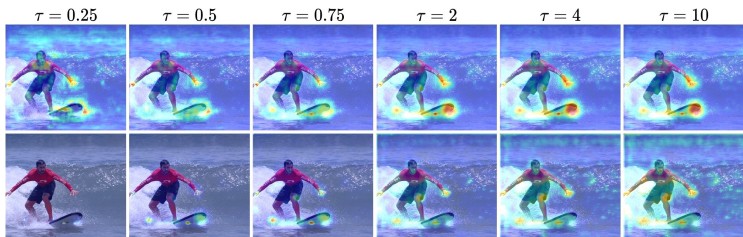

Figure 4: Distribution of the distillation loss for KL (*tempered softmax*, top) and TLD (*tempered sigmoid*, bottom) by varying $\tau$. Best viewed with zoom-in.

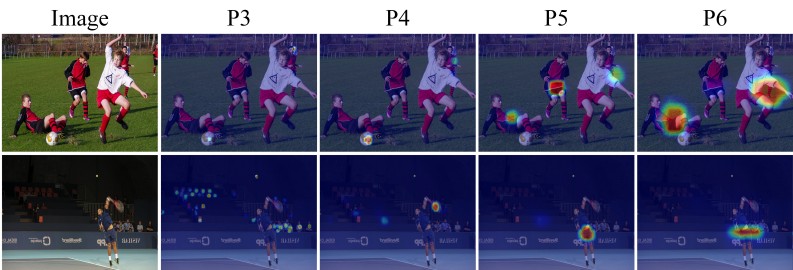

Figure 5: Visualization of loss distribution in different FPN levels with $\tau = 0.5$. P3∼P6 represent the FPN levels. The teacher-student pair is GFocal with ResNet101-ResNet18 as the backbones. Best viewed with zoom-in.

performance promotion. For instance, with ResNet34 as the backbone, the student gets +2.54% and +1.1% absolute mAP improvements in the classification and detection tasks, respectively.

**Loss Distribution on Multi-Level.** Here, we give details on loss distributions at different levels of feature pyramid networks (FPN) (Lin et al., 2017a) in the detection task. As illustrated in Figure 5, the beneficial foreground regions provoke higher loss, while the meaningless background gets negligible weights. More importantly, the loss in different FPN layers has different regions of interest. The low-level FPN is more concentrated on the tiny objects (even the extremely small *sports ball*, see P3), while the high-level FPN level pays more attention to the large instances. This property effectively relieves the redundancy in the multi-level KD and leads to balanced detection improvements. More visualizations can be found in Figure B.3.

Please refer to Appendix for additional experimental results and relevant discussions.

## 5 CONCLUSION

In this paper, we investigated knowledge distillation under multi-label learning. Our empirical results clarify that the vanilla *sigmoid* and *tempered softmax* are both inferior in performing distillation in multi-label learning. As a simple yet effective solution, we introduced *tempered sigmoid* and proposed the tempered logit distillation (TLD). We provided theoretical and visual justification, showing that our method realizes hard mining during distillation, which is the primary attribute of its success. Our TLD is general and achieves outstanding distillation results in various vision tasks, including image classification, object detection, and instance segmentation. Only distilling with the logit, TLD even outperforms state-of-the-art KD methods designed specifically for the corresponding tasks. Besides, we notice some quite interesting observations regarding the temperature $\tau$, which may supplement and consummate its behavior in logit distillation.

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

## A EXPERIMENTAL SETTINGS

### A.1 DATASETS

COCO. COCO (Lin et al., 2014), covering 80 categories, is the standard benchmark for classification, detection, and instance segmentation tasks. COCO contains 115K and 5K images for training and validation. For image classification, we use COCO-2014 following L2D (Yang et al., 2023b). COCO-2017 is used for detection and instance segmentation as the common practice.

PASCAL-VOC. PASCAL-VOC (Everingham et al., 2015) consists of 5K and 4K images, across 20 classes, for training and validation.

NUS-WIDE. NUS-WIDE (Chua et al., 2009), which is a large-scale dataset commonly used for image classification with 81 concept categories, includes 161K and 107K annotated images for training and validation, respectively.

### A.2 EVALUATION METRICS

For classification, we choose the mean Average Precision (mAP), overall F1-score (OF1), and average per-category F1-score (CF1) as the metrics. For object detection and instance segmentation, we report mAP as the main metric, together with AP under different IoU thresholds $AP_{50}$, $AP_{75}$ and object scales $AP_S$, $AP_M$, $AP_L$.

### A.3 NETWORK ARCHITECTURES

For backbones, we consider several models: ResNet (He et al., 2016), WRN (Zagoruyko & Komodakis, 2016), RepVGG (Ding et al., 2021), MobileNet (Sandler et al., 2018). Besides, vision transformer-based networks, such as Swin (Liu et al., 2021), are also included in the image classification task. For the detection task, various dense detectors are selected, i.e., RetinaNet (Lin et al., 2017b), FCOS (Tian et al., 2019), ATSS (Zhang et al., 2020), GFocal (Li et al., 2020), and RepPoints (Yang et al., 2019). For the instance segmentation task, we choose SOLOV2 (Wang et al., 2020). All the backbones are pre-trained on ImageNet (Deng et al., 2009).

### A.4 TRAINING DETAILS

For image classification, we use the L2D codebase (Yang et al., 2023b). Specifically, we respectively train the teachers and students for 30 and 80 epochs with the Adam optimizer (Kingma & Ba, 2015). The one-cycle policy is used with a maximal learning rate of 1e-4 and a weight decay of 1e-4. The batch size is 64 and the input size is 224×224. For each training image, we apply a weak augmentation consisting of random horizontal flipping and a strong augmentation consisting of Cutout (DeVries & Taylor, 2017) and RandAugment (Cubuk et al., 2020). We fix the $\tau = 0.75$ and tune the $\lambda$ in a reasonable range.

For object detection and instance segmentation, our implementation is built upon the MMDetection (Chen et al., 2019a) framework with default configure files. All the *student* models are trained under the 1× learning schedule without any tricks, e.g., multi-scale training. We fix the $\tau = 0.5$ and tune the $\lambda$ in a reasonable range.

### A.5 COMPARED METHODS

The references of the compared methods in image classification are PKT (Passalis & Tefas, 2018), RKD (Park et al., 2019), Review KD (Chen et al., 2021), PS (Song et al., 2021), MSE (Xu et al., 2022), and L2D (Yang et al., 2023b).

The references of the compared methods in object detection and instance segmentation are FitNet (Adriana et al., 2015), FGFI (Wang et al., 2019), DeFeat (Guo et al., 2021), CWD (Shu et al., 2021), GID (Dai et al., 2021), FGD (Yang et al., 2022b), MGD (Yang et al., 2022c), PKD (Cao et al., 2022), DIST (Huang et al., 2022), SSKD (De Rijk et al., 2022), RM (Li et al., 2022a), LD (Zheng et al., 2022), MasKD (Huang et al., 2023), and BCKD (Yang et al., 2023a).

Table B.1: Impact of loss weight hyper-parameter $\lambda$ on GFocal ResNet101-ResNet18 with $\tau = 0.5$.

| $\lambda$ | 0.5 | 1.0 | 1.5 | 2.0 | 2.5 | 3.0 |
|---|---|---|---|---|---|---|
| mAP | 36.6 | 37.2 | 37.1 | 37.1 | **37.3** | 36.9 |
| $AP_{50}$ | 54.1 | 55.2 | 55.2 | 55.1 | **55.4** | 55.2 |
| $AP_{75}$ | 39.4 | 40.0 | 39.8 | 39.7 | **40.1** | 39.7 |

Table B.2: Multi-Label image classification KD performance on COCO.

| Method | ResNet101-MobileNetV2 | | | SwinTiny-ResNet34 | | | ResNet101-RepVGGA0 | | | SwinSmall-SwinTiny | | |
|---|---|---|---|---|---|---|---|---|---|---|---|---|
| | mAP | OF1 | CF1 | mAP | OF1 | CF1 | mAP | OF1 | CF1 | mAP | OF1 | CF1 |
| Teacher | 73.62 | 73.89 | 68.61 | 79.43 | 78.77 | 75.07 | 73.62 | 73.89 | 68.61 | 81.70 | 80.48 | 77.12 |
| Student | 71.85 | 73.68 | 68.40 | 70.31 | 72.49 | 66.82 | 70.02 | 72.49 | 66.77 | 79.59 | 79.18 | 75.42 |
| RKD | 71.76 | 73.68 | 68.40 | 70.00 | 72.34 | 66.64 | 70.08 | 72.35 | 66.72 | 79.59 | 79.18 | 75.42 |
| PKT | 71.88 | 73.60 | 68.35 | 69.99 | 72.35 | 66.56 | 70.11 | 72.47 | 66.80 | 79.64 | 79.09 | 75.39 |
| Review KD | 71.92 | 73.73 | 68.48 | 70.29 | 72.39 | 66.58 | 70.00 | 72.33 | 66.62 | 79.81 | 79.18 | 75.55 |
| MSE | 71.91 | 73.68 | 68.28 | 70.33 | 72.57 | 66.72 | 70.07 | 72.50 | 66.85 | 79.67 | 79.20 | 75.52 |
| PS | 72.11 | 73.89 | 68.42 | 70.94 | 72.93 | 67.57 | 70.30 | 72.61 | 67.10 | 79.96 | 79.64 | 76.20 |
| BCE | 72.17 | 73.84 | 68.52 | 71.14 | 72.99 | 67.63 | 70.48 | 72.77 | 67.10 | 80.11 | 79.68 | 76.44 |
| L2D | 73.17 | 74.71 | 69.37 | 72.39 | 74.15 | 68.63 | 72.01 | 73.99 | 68.58 | 80.86 | 80.36 | 77.20 |
| TLD | **73.40** | **74.80** | **69.52** | **73.98** | **75.23** | **70.34** | **72.91** | **74.50** | **69.26** | **82.24** | **81.19** | **78.30** |
| BCE + L2D | 73.24 | 74.85 | 69.72 | 73.42 | 74.97 | 70.20 | 72.14 | 74.08 | 68.78 | 81.59 | 81.03 | 77.86 |
| TLD + L2D | **74.40** | **75.59** | **70.31** | **75.04** | **76.08** | **71.18** | **73.84** | **75.13** | **69.81** | **83.05** | **81.95** | **78.91** |

# B ADDITIONAL EXPERIMENTS AND VISUALIZATIONS

## B.1 SENSITIVITY OF $\lambda$

Here, we perform additional experiments to study the impact of $\lambda$ by setting $\tau = 0.5$ with the detection task. As shown in Table B.1, TLD achieves stable performance with $\lambda$'s range in $[1.0, 3.0]$, demonstrating that our method is not sensitive to the choice of $\lambda$. In practice, the loss weight is suggested to keep a similar amount loss value of the task loss in the classification head (Zhao et al., 2022; Hinton et al., 2015; Huang et al., 2022).

## B.2 IMAGE CLASSIFICATION

Table B.2 and Table B.3 summarize more KD results on COCO (Lin et al., 2014) and PASCAL-VOC (Everingham et al., 2015), respectively. It is observed that our TLD consistently surpasses the previous methods on both homogeneous and heterogeneous KD pairs, verifying its effectiveness.

Figure B.1 plots the class-wise AP scores. It is shown that our method consistently achieves higher results than L2D (Yang et al., 2023b) in most categories, showing that the performance gains of the proposed approach are holistic.

Table B.3: Multi-Label image classification KD performance on PASCAL-VOC.

| Method | ResNet50-RepVGGA0 | | | SwinTiny-ResNet18 | | | ResNet50-MobileNetV2 | | | SwinTiny-MobileNetV2 | | |
|---|---|---|---|---|---|---|---|---|---|---|---|---|
| | mAP | OF1 | CF1 | mAP | OF1 | CF1 | mAP | OF1 | CF1 | mAP | OF1 | CF1 |
| Teacher | 86.73 | 84.92 | 81.21 | 91.43 | 89.81 | 87.63 | 86.73 | 84.92 | 81.21 | 91.43 | 89.81 | 87.63 |
| Student | 83.79 | 83.36 | 79.83 | 84.01 | 83.60 | 79.42 | 86.12 | 85.01 | 81.76 | 86.12 | 85.01 | 81.76 |
| RKD | 84.26 | 84.29 | 80.70 | 83.27 | 83.05 | 79.55 | 86.22 | 84.97 | 81.76 | 85.68 | 85.31 | 81.57 |
| PKT | 83.93 | 83.79 | 80.03 | 83.45 | 83.25 | 79.64 | 86.10 | 84.84 | 81.66 | 85.67 | 85.22 | 81.68 |
| Review KD | 84.07 | 83.62 | 80.34 | 83.37 | 83.08 | 78.93 | 85.87 | 85.04 | 81.73 | 85.69 | 85.10 | 81.56 |
| MSE | 84.01 | 84.05 | 80.52 | 83.60 | 83.06 | 79.46 | 86.20 | 84.94 | 81.84 | 85.80 | 85.51 | 81.98 |
| PS | 84.80 | 84.46 | 81.13 | 83.97 | 83.75 | 79.86 | 86.26 | 85.47 | 82.06 | 86.07 | 85.73 | 82.39 |
| BCE | 85.07 | 84.91 | 81.55 | 84.61 | 84.26 | 80.78 | 86.38 | 85.67 | 82.43 | 86.11 | 85.98 | 82.55 |
| L2D | 85.94 | 85.50 | 82.22 | 85.10 | 85.09 | 81.36 | 86.91 | 85.39 | 82.03 | 86.97 | 86.26 | 83.12 |
| TLD | **86.51** | **85.64** | **82.38** | **85.80** | **85.29** | **81.77** | **87.36** | **86.35** | **83.31** | **87.13** | **86.54** | **83.34** |
| BCE + L2D | 86.26 | 85.85 | 82.55 | 85.87 | 85.67 | 82.17 | 87.32 | 86.48 | 83.26 | 87.37 | 86.88 | 83.68 |
| TLD + L2D | **86.64** | **85.48** | **82.21** | **86.48** | **85.50** | **82.47** | **87.72** | **86.49** | **83.35** | **88.00** | **87.09** | **84.04** |

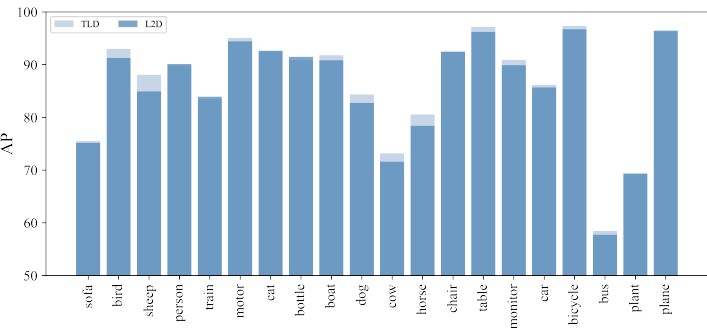

Figure B.1: Per-class AP on PASCAL-VOC (Everingham et al., 2015). Best viewed with zoom-in.

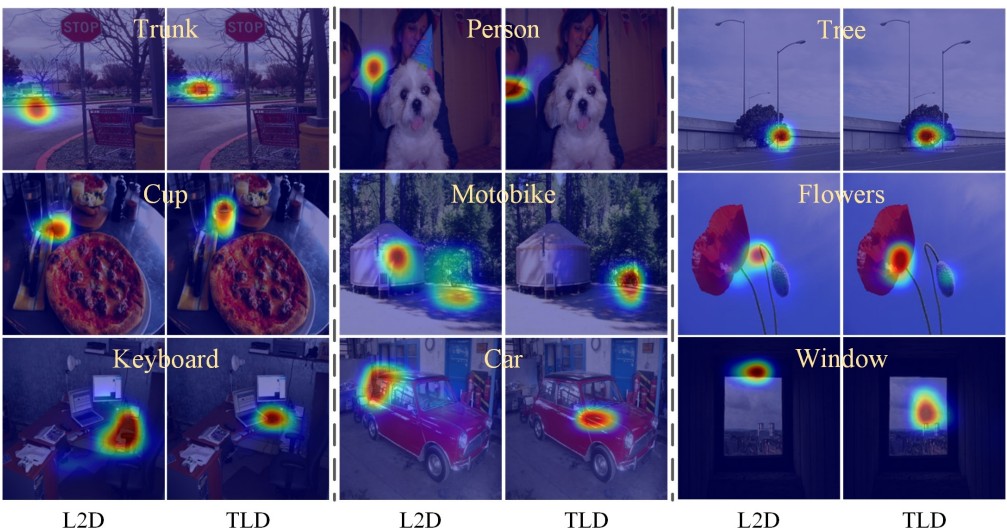

L2D          TLD          L2D          TLD          L2D          TLD

Figure B.2: More visualizations of the class activation maps generated from L2D and our TLD (from left to right: COCO (Lin et al., 2014), PASCAL-VOC (Lin et al., 2014), NUS-WIDE (Chua et al., 2009)). The colored texts are the query categories. Best viewed in color with zoom-in.

Figure B.2 shows more visualizations of class activation maps. We can see that our TLD can locate the specified objects more precisely than the L2D (Yang et al., 2023b), which directly transfers to performance enhancement.

### B.3 OBJECT DETECTION

Table B.4 gives more KD results on object detection. It is observed that our method consistently boots the performance of various detectors, demonstrating its generalization and versatility.

In Figure B.3, we visualize more loss distribution in the logit space over different FPN levels. One can see that our method successfully recognizes worthy pixels for targets, regardless of the instance scales and categories. An interesting phenomenon is that the useful KD region for an instance is generally smaller than the whole instance. A similar observation can be drawn from (Huang et al., 2023), which performs the distillation with the feature.

## C THEORETICAL ANALYSIS

### C.1 EXPLANATION ABOUT THE GRADIENT IN EQ. (8)

In our method, we first reduce the multi-label classification task into a set of binary classification tasks, then we distill each binary classification independently with KL-divergence. Formally, the

Table B.4: Distillation performance of various detectors on COCO.

| Method | mAP | AP$_{50}$ | AP$_{75}$ | AP$_S$ | AP$_M$ | AP$_L$ |
|---|---|---|---|---|---|---|
| ATSS-R101 | 41.5 | 59.9 | 45.2 | 24.2 | 45.9 | 53.3 |
| ATSS-R50 | 39.4 | 57.6 | 42.8 | 23.6 | 42.9 | 50.3 |
| TLD | **41.2** | **59.9** | **44.4** | **24.8** | **45.1** | **52.2** |
| FCOS-R101 | 39.1 | 58.3 | 42.1 | 22.7 | 43.3 | 50.3 |
| FCOS-R50 | 36.6 | 56.0 | 38.8 | 21.0 | 40.6 | 47.0 |
| TLD | **37.9** | **58.0** | **40.2** | **22.1** | **42.0** | **48.5** |
| Rep-R101 | 40.5 | 61.3 | 43.5 | 23.4 | 44.7 | 52.2 |
| Rep-R50 | 38.1 | 58.7 | 40.8 | 22.0 | 41.9 | 50.1 |
| TLD | **39.6** | **61.2** | **42.4** | **24.2** | **43.7** | **51.3** |
| Retina-R101 | 38.9 | 58.0 | 41.5 | 21.0 | 42.8 | 52.4 |
| Retina-R50 | 36.5 | 55.4 | 39.1 | 20.4 | 40.3 | 48.1 |
| TLD | **38.4** | **58.3** | **41.0** | **21.9** | **42.2** | **50.5** |

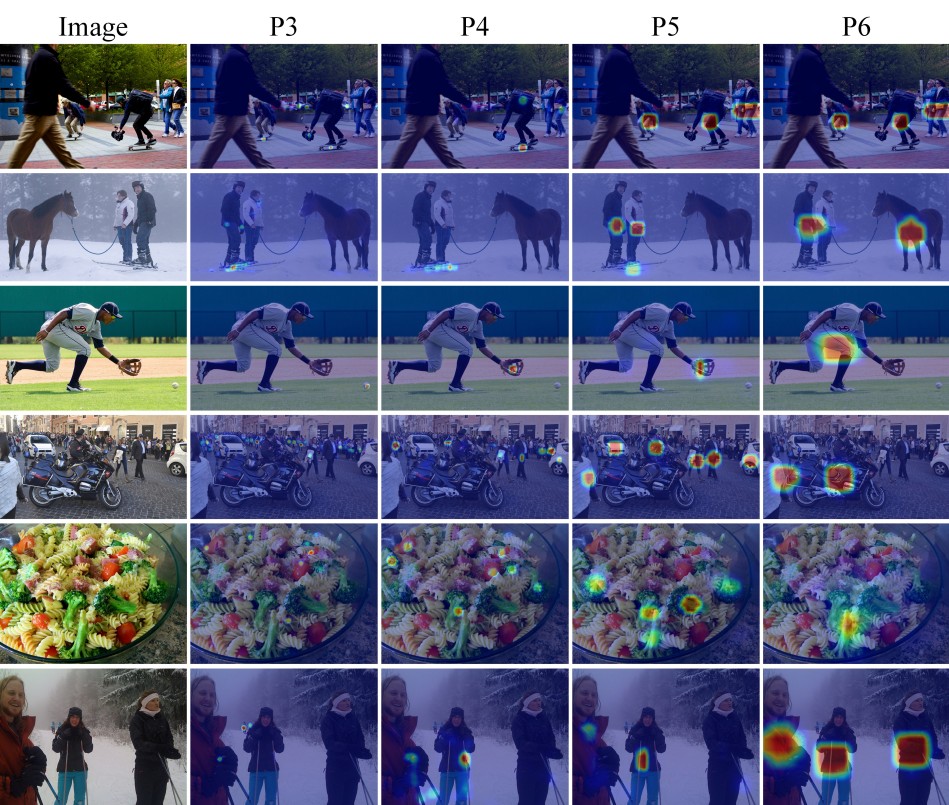

Figure B.3: More visualizations of the loss distribution in different FPN levels with $\tau = 0.5$. P3~P6 represent the FPN levels. The KD pair is GFocal with ResNet101-ResNet18 as the backbones. Best viewed in color with zoom-in.

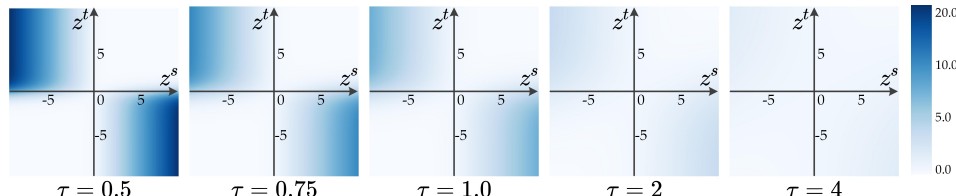

Figure C.4: The value of $\mathcal{L}_{\text{TLD}}$ by varying the temperature $\tau$. A darker color indicates a higher loss.

prediction $\mathbf{p}_\tau \in \mathbb{R}^{C \times 1}$ is converted to $\tilde{\mathbf{p}}_\tau = [[\mathbf{p}_{1,\tau}, 1 - \mathbf{p}_{1,\tau}], ..., [\mathbf{p}_{c,\tau}, 1 - \mathbf{p}_{c,\tau}]] \in \mathbb{R}^{C \times 2}$, where $\mathbf{p}_{i,\tau} = \frac{1}{1+e^{-z_i/\tau}}$. $z_i$ is the logit, $i = 1, ..., C$. We distill the $\tilde{\mathbf{p}}_\tau$ with KL divergence in a per-class manner as:

$$\mathcal{L}_{\text{TLD}} = \mathcal{L}_{\text{KL}}(\tilde{\mathbf{p}}_{i,\tau}^s, \tilde{\mathbf{p}}_{i,\tau}^t) \tag{9}$$

where $\tilde{\mathbf{p}}_{i,\tau}^s = [\mathbf{p}_{i,\tau}^s, 1 - \mathbf{p}_{i,\tau}^s]$ and $\tilde{\mathbf{p}}_{i,\tau}^t = [\mathbf{p}_{i,\tau}^t, 1 - \mathbf{p}_{i,\tau}^t]$. Then, we can derive the standard KL gradient as show in Eq. (8).

### C.2 REFORMULATION $\mathcal{L}_{\text{TLD}}$

Here, we further theoretically explore the impact of temperature scaling in our TLD. Recall that, our method distills per-class independently, so we take one class as an example to ease the presentation. Extending it to the multi-class case is straightforward. First, we have $\tilde{\mathbf{p}}_\tau = [\mathbf{p}_\tau, 1 - \mathbf{p}_\tau]$, where:

$$\mathbf{p}_\tau = \frac{1}{1 + e^{-z/\tau}}, \quad 1 - \mathbf{p}_\tau = 1 - \frac{1}{1+e^{-z/\tau}} = \frac{e^{-z/\tau}}{1 + e^{-z/\tau}} = e^{-z/\tau} \cdot \mathbf{p}_\tau \tag{10}$$

, and the $z$ is the logit. Then, based on Eq. (10), the $\mathcal{L}_{\text{TLD}}$ can be expressed as follows:

$$
\begin{aligned}
\mathcal{L}_{\text{TLD}} &= -(\mathbf{p}_\tau^t \log \mathbf{p}_\tau^s + (1 - \mathbf{p}_\tau^t) \log(1 - \mathbf{p}_\tau^s)) \\
&= -(\mathbf{p}_\tau^t \log \mathbf{p}_\tau^s + (1 - \mathbf{p}_\tau^t) \log(e^{-z^s/\tau} \cdot \mathbf{p}_\tau^s)) \\
&= -(\mathbf{p}_\tau^t \log \mathbf{p}_\tau^s + (1 - \mathbf{p}_\tau^t)(\log \mathbf{p}_\tau^s - z^s/\tau)) \\
&= -(\mathbf{p}_\tau^t \log \mathbf{p}_\tau^s + (1 - \mathbf{p}_\tau^t) \log \mathbf{p}_\tau^s) + (1 - \mathbf{p}_\tau^t)(z^s/\tau) \\
&= -\log \mathbf{p}_\tau^s + (1 - \mathbf{p}_\tau^t)(z^s/\tau) \\
&= \log(1 + e^{-z^s/\tau}) + \frac{e^{-z^t/\tau}}{1 + e^{-z^t/\tau}}(z^s/\tau)
\end{aligned}
\tag{11}
$$

In Figure C.4, we visualize the value of $\mathcal{L}_{\text{TLD}}$ by varying the temperature $\tau$. It is clear that our method with $\tau < 1$ is more sensitive to hard samples and will regulate the student to pay more learning effort to them.

