# OpenReview forum: "Unleashing the Potential of Temperature Scaling for Multi-Label Logit Distillation"
_ICLR.cc/2025/Conference — ICLR 2025 Conference Withdrawn Submission_

### Official Review · Reviewer_mZnK · 2024-10-21

**Soundness:** 3
**Presentation:** 3
**Contribution:** 2
**Rating:** 6
**Confidence:** 4

**Summary:**

This paper presents a novel approach to multi-label logit distillation by introducing a tempered sigmoid activation function with temperature scaling. The authors argue that both the traditional sigmoid and tempered softmax activation functions have intrinsic limitations in multi-label distillation. To address these issues, they propose Tempered Logit Distillation (TLD), a method that modifies the sigmoid activation function with temperature scaling, offering significant performance improvements across multi-label learning tasks such as image classification, object detection, and instance segmentation. Comprehensive experiments demonstrate the effectiveness of TLD on benchmarks like COCO, PASCAL-VOC, and NUS-WIDE, showing consistent gains in performance.

**Strengths:**

1. Tempered Sigmoid is novel and addresses key limitations in existing multi-label logit distillation methods.

2. The paper provides comprehensive experimental results, including a well-designed ablation study, which thoroughly evaluates the contribution of each component.

3. The proposed method achieves state-of-the-art (SOTA) performance on multiple benchmarks, demonstrating its effectiveness and broad applicability in multi-label learning tasks.

**Weaknesses:**

1. **Weak theoretical foundation**: The theoretical analysis in the paper primarily revolves around the choice of the hyperparameter $\tau$, but the reasoning behind different $\tau$ values lacks depth and conviction. While the paper presents justifications for the selection of $\tau$, it does not offer strong theoretical support that would convincingly validate these choices.

2. **Weak motivation**: The motivation provided in the introduction feels somewhat forced. For example, the second point in the introduction's analysis seems weaker because the results in the table show that sigmoid performs quite well when combined with common tricks, which undermines the strength of the argument against sigmoid.

3. **Method simplicity**: The proposed method can be viewed as a generalized version of sigmoid by simply adding a controllable temperature hyperparameter. As a reviewer, I generally appreciate methods that are simple but effective. However, this approach seems bound to outperform previous methods because it inherently expands the search space to include prior approaches, which raises questions about its novelty.

4. **Fairness of comparisons**: Given the point raised in 3, the method can be seen as akin to adding a regularization term. A more fair comparison would involve adding analogous regularization to competing methods, rather than purely comparing methods without such adjustments. More meaningful comparisons would include integrating this method with more tricks commonly used in the field, such as multi-scale training (as mentioned in the appendix). Demonstrating that this method can provide orthogonal improvements when combined with other techniques would make the contribution more significant.

**Questions:**

1. In classification tasks, the student model outperforms the teacher, as shown in Table 1. Is this improvement due to the benefits of knowledge distillation (such as self-distillation), or could it be attributed to the training paradigm, where the teacher's paradigm may not be as suitable for the task as the student's? What would be the effect of adapting the teacher to better fit the task?

2. In Table 10, the impact of $\tau$ on KL divergence is not fully explored. What happens to the method's performance when $\tau$ is closer to 4?

3. Different values of $\tau$ were used for classification and segmentation. Besides running all possible $\tau$ values, is there a more insightful way to analyze and determine the optimal $\tau$? In previous work with softmax, was τ treated as fixed or variable?

---

> ### Author Response · Authors · 2024-11-18
> **Response to Reviewer mZnK**
>
> Thank you for your positive comments and efforts in reviewing our paper. The responses to your comments and questions are as follows.
> Please let us know if you still have any unclear part of our work. We are actively available during this rebuttal!
>
>
> **Q1**: Weak theoretical foundation: The theoretical analysis behind different τ values lacks depth and conviction.
> **A1**: In the manuscript, we provided the theoretical justification in the gradient space. We further give analysis by reformulating our distillation loss in Appendix C.2. With the loss reformulation and new visualization (Figure C.4), the conclusion is aligned with the one drawn in the manuscript that our method with $\tau<1$ will regulate the student to stress the distillation on the hard samples.
> ***
> **Q2**: Weak motivation: Some KD results undermine the strength of the argument against sigmoid.
> **A2**: We hope the following logical flow may highlight our motivation and address your concern accordingly.
> (A) Tempered softmax is not better than vanilla sigmoid: The prior publications [1][2] claimed that the vanilla sigmoid per se is better than the tempered softmax in multi-label logit KD. However, the empirical evidence reported in their paper is based on combining the vanilla sigmoid with other tricks (e.g., [2] used the distillation mask). From Table 1, we can see that the vanilla sigmoid per se actually obtains similar or even worse KD results as tempered softmax when the tricks are removed. These results imply: (A.1) the argument in [1][2] is not entirely correct and we clarify it (one of our contributions); (A.2) the vanilla sigmoid per se and tempered softmax are both empirically inferior in performing KD for multi-label learning tasks.
>
> (B) Sigmoid should be better than softmax: The softmax can capture and transfer the inter-class relations (dark knowledge [3]) from the teacher to student, which is the primary attribute of its success in KD [3]. However, the class is individually activated in multi-label learning, which violates the attribution of softmax. In other words, the softmax (even with temperature scaling) is naturally sub-optimal for multi-label logit KD, as shown in Tables 1,6,9,10. Sigmoid, on the other hand, aligns the classifier behavior of multi-label learning, i.e., the sigmoid should be better than softmax. As you mentioned, the sigmoid performs quite well when combined with an advanced loss function (i.e., DIST [4]). In this regard, these results actually strengthen our motivation and argument. Please note that these results do not affect the conclusions of (A.1) and (A.2). Besides, as shown in our response for Q4: Fairness of comparisons, the proposed tempered sigmoid works better with DIST.
>
> (C) What limits sigmoid: Given (A.2) and (B), we wonder what limits the power of sigmoid in multi-label KD. Our answer is the vanilla sigmoid ignoring the critical temperature scaling operation in KD. To validate this conjecture, our work gives empirical and theoretical evidence.
>
> [1] Multi-label Knowledge Distillation, 2023, ICCV.
> [2] Bridging Cross-task Protocol Inconsistency for Distillation in Dense Object Detection, 2023, ICCV.
> [3] Distilling the Knowledge in a Neural Network, 2014, NeurIPS Workshop.
> [4] Knowledge Distillation from A Stronger Teacher, 2022, NeurIPS.

---

> > ### Author Response · Authors · 2024-11-18
> > **Response to Reviewer mZnK**
> >
> > **Q3**: Method simplicity.
> > **A3**: We totally agree that the temperature scaling is a simple operation in knowledge distillation. However, simplicity does not negate its importance and practical benefits. As discussed in Sec. 1 and Sec. 2, temperature scaling actually plays a pivotal role in logit-based distillation and has been actively studied for boosting KD [5][6][7][8] (More papers based on temperature scaling have been cited in Sec. 2). Unfortunately, all these methods are built on softmax in single-label learning. As far as we know, the behavior of temperature scaling in sigmoid for multi-label logit distillation has not been probed. In this paper, we investigate three under-exploited questions (please refer to the introduction section) and provide clear answers. In particular, (1) we clarify that a recently discovered perspective the vanilla sigmoid is superior to tempered softmax is not entirely correct; (2) we disclose ignoring is the essential reason causing the vanilla sigmoid to perform poorly; (3) we find that the temperature behavior in multi-label learning ($\tau<1$) and single-label learning ($\tau>1$) is exactly opposite. Our work consummates the behavior of temperature scaling in logit KD and showcases valuable insights for its wide application and future research. Thus, the overall contributions and novelty are justified. We sincerely hope you may shift the attention from simplicity to the contributions and effectiveness of the proposed method.
> >
> > [5] Asymmetric Temperature Scaling Makes Larger Networks Teach Well Again, 2022, NeurIPS.
> > [6] Curriculum Temperature for Knowledge Distillation, 2023, AAAI.
> > [7] Logit Standardization in Knowledge Distillation, 2024, CVPR.
> > [8] Knowledge Distillation Based on Transformed Teacher Matching, 2024, ICLR.
> > ***
> > **Q4**: Fairness of comparisons: adding analogous regularization to competing methods and integrating our method with more tricks.
> > **A4**: We would like to list, with all due respect, some acknowledged facts in KD before addressing your concern.
> > (1) KD is categorized as feature-based (i.e., distilling with the feature map) and logit-based (i.e., distilling with the logit), and our method is a pure logit-based one.
> > (2) The temperature scaling is used in the activation functions, which are only involved in the logit-based KD methods.
> > (3) The feature-based methods are naturally superior to logit-based, especially for dense tasks.
> > (4) Feature-based and logit-based methods can be combined for distilling the students.
> > (5) All the compared methods are feature-based, except KL, BCE, CWD, and DIST.
> >
> > Based on the above facts, we make the following clarifications:
> > A: With (1)(2)(5), the proposed methods can not be integrated with most of the compared methods.
> > B: With (1)(4), we have already included related results in Tables 2, 3, 4, 8, 9. The results show that the students indeed gain extra and orthogonal improvements.
> > C: With (1)(3), the main KD results in the paper validate the effectiveness of the proposed method.
> > D: KL and BCE are the two competitors of our method.
> > C: The original CWD activates the logit map with tempered softmax along the channel-wise. However, tempered sigmoid is an element-wise activation function, i.e., there is no difference between activating the logit map along the channel-wise or spatial-wise. In other words, CWD using tempered sigmoid is formally equivalent to our TLD and it would be no meaning to combine CWD and ours.
> > D: Our method is applicable to DIST. In our paper, we actually have reported corresponding KD results (see Table 2), where further improvement is achieved, verifying the proposed TLD could provide orthogonal improvements when combined with other techniques.
> > F: The multi-scale training (mentioned in the appendix) is a special training configuration for detectors, not a KD trick/method.
> > Therefore, the comparisons in our paper are generally fair and meaningful.

---

> ### Author Response · Authors · 2024-11-18
> **Response to Reviewer mZnK**
>
> **Q5**: It is the improvement due to the benefits of KD or could it be attributed to the training paradigm, where the teacher's paradigm may not be as suitable for the task as the student's?
> **A5**: We point out that (1) all the undistilled students are inferior to the pre-trained teacher; (2) the teachers have identical training paradigms (including data augmentation, batch size, input size, learning rate, optimizer, see Appendix A) with the undistilled/distilled students, except the training epoch; (3) we empirically observer that the prolonged training actually degrades the teacher accuracy due to overfitting; (4) the setting of teachers’ training epoch is also followed by [1] for a fair comparison across KD methods. This said, the performance improvement of students certainly comes from knowledge distillation rather than a better training paradigm.
> ***
> **Q6**: What would be the effect of adapting the teacher to better fit the task?
> By ‘task’ here,
> **(1)** if you refer to the image classification task:
> **A6.1**: For sure, one can adopt or tune a high-accuracy teacher to distill the students, in which case the distilled student may still be inferior to the teacher. Together with Q5 and A5, we also clarify that KD does not guarantee the distilled students consistently outperform the teachers in any task (i.e., classification, detection, and segmentation). Since the ultimate goal of KD is to obtain performant students, it might be better to focus on the comparison of the distilled student’s performance between different KD methods, who share exact training/KD settings.
>
> **(2)** if you refer to the knowledge distillation task, i.e., adapting the teacher to distill better students:
> **A6.2**: This is an insightful thought! As a matter of fact, obtaining a “good” teacher for distillation is an active topic (e.g.,[9][10][11]) in KD. Note that, all these methods are limited to the single-label classification task. Please understand that it might be challenging to conceive a brand-new KD method for a “good” teacher in multi-label learning tasks within the rebuttal period. Besides, in our paper, we actually have conducted a similar experiment, where the teacher is adapted by varying the mode size (Sec 4.1, Table 5). It is observed that a high-accuracy teacher tends to distill a better student with our method.
>
> [9] Learning Student-Friendly Teacher Networks for Knowledge Distillation, 2021, NeurIPS.
> [10] Toward Student-Oriented Teacher Network Training for Knowledge Distillation, 2024, ICLR.
> [11] How to Train the Teacher Model for Effective Knowledge Distillation, 2024, ECCV.
> ***
> **Q7**: What happens to the method’s performance when the temperature is closer to 4.
> **A7**: The below table gives additional results (i,e., more value of $\tau$) on classification over COCO. The teacher-student pair is ResNet101-ResNet34, and mAP is reported. It is observed that the performance trend is clearly aligned with our findings: (1) the tempered softmax delivers better results when $\tau>1$ while our tempered sigmoid works with $\tau<1$; (2) our method is superior to tempered softmax in multi-label logit KD.
> | $\tau$           | 0.25  | 0.4   | 0.5   | 0.6   | 0.75  | 2.0   | 3.0   | 4.0   | 5.0   | 6.0   |
> | ---------------- | ----- | ----- | ----- | ----- | ----- | ----- | ----- | ----- | ----- | ----- |
> | Tempered Softmax | 70.69 | 70.57 | 71.12 | 70.73 | 71.38 | 72.33 | 72.41 | 72.57 | 72.47 | 72.40 |
> | Tempered Sigmoid | 73.14 | 73.40 | 73.48 | 73.34 | 73.26 | 71.68 | 72.01 | 70.68 | 71.09 | 71.10 |
> ***
> **Q8**: Different values of temperature used for classification and segmentation.
> **A8**: Yes, we assign a lower temperature value ($\tau=0.5$) for the dense prediction tasks, i.e., object detection and instance segmentation. However, as shown in Table 10, $\tau=0.75$ only leads to 0.1 AP drop for dense prediction tasks compared with $\tau=0.5$. $\tau=0.75$ is used in the classification task. We ascribe this setting to the fact that the dense prediction tasks contain massive easy and futile background areas. In this case, a lower $\tau$ can regulate the students to tilt distilling the informative foreground, as shown in Figure 4.

---

> ### Author Response · Authors · 2024-11-18
> **Response to Reviewer mZnK**
>
> **Q9**: Besides running all possible $\tau$ values, is there is a more insightful way to analyze the determine the optimal temperature?
> **A9**: Indeed, there are some insightful ways for determine the value of $\tau$ in our TLD. For example, as the response in Q8, a lower temperature favors the challenging dense prediction tasks, which contain massive easy background pixels. We also conjecture the gap between the teacher and student could be reliable guidance for tuning $\tau$. Intuitively, the smaller the gap is, the more samples can be easily learned by the student. Namely, the smaller $\tau$ should be applied to increase the gradient contributed by the hard samples. The below table gives empirical evidence on COCO. mAP is reported.
> | $\tau$ | ResNet101-ResNet18 | ResNet34-ResNet18 |
> | ------ |:------------------: |:-----------------: |
> | 0.25   | 69.02             | 70.02             |
> | 0.5    | 70.26             | 70.03           |
> | 0.75   |         71.19           |              69.68     |
> ***
> **Q10**: In previous work with softmax, was $\tau$ treated as fixed or variable?
> **A10**: The temperature $\tau$ is variable and, most importantly, adjusting it brings noticeable practical benefits. Please refer to our answer for Q3 and the discussion of related work (Sec. 2) for more information.

---

> > ### Comment · Reviewer_mZnK · 2024-11-19
> >
> > Thank you very much for your detailed response and clarification.
> >
> > I will maintain my current positive score and increase my confidence.

---

> > > ### Author Response · Authors · 2024-11-20
> > > **Thank you for following up!**
> > >
> > > Thank you so much for the feedback and considering rasing the confidence! If you have any further questions, let us know any time!

---

### Official Review · Reviewer_dFkV · 2024-10-28

**Soundness:** 3
**Presentation:** 2
**Contribution:** 2
**Rating:** 6
**Confidence:** 3

**Summary:**

This paper introduces a new Knowledge Distillation (KD) approach for multi-label learning, that aims at distilling the knowledge from the teacher model to a student model. The proposed approach, named Tempered Logit Distillation (TLD), is simple and uses a sigmoid with a temperature scaling to modulate the logit smoothness. The tempered sigmoid is applied on the model outputs, so it seems easy to use in practice. Then, the Kullback-Leibler (KL) divergence loss is applied on the sigmoid outputs to distill the knowledge from the teacher model to a student model. The proposed approach shows good performances on three standard datasets: COCO, PASCAL VOC, and NUS-WIDE. The paper contains results for several model architectures, and several tasks: image classification, object detection, and instance segmentation. The paper also contains an ablation analysis of the proposed approach.

**Strengths:**

- The paper introduces the Tempered Logit Distillation (TLD), a knowledge distillation loss for multi-label classification. First, a sigmoid with a temperature scaling is applied on logit outputs. Then, the Kullback-Leibler (KL) divergence loss is applied on the sigmoid outputs to distill the knowledge from the teacher model to a student model. The proposed Knowledge Distillation (KD) method is simple and seems easy to use in practice as it requires access to the model outputs only. It is not necessary to access intermediate feature maps.
- The proposed approach is evaluated on three standard multi-label classification benchmarks: COCO, PASCAL VOC, and NUS-WIDE. The results with multiple model architectures show that the TLD outperforms existing approaches.
- The paper shows that TLD works well on different tasks like object detection and instance segmentation.
- The paper contains multiple analyses of the proposed approach. Table 1 shows an ablation study. Table 5 shows the relation between the student and teacher. Table 6 analyzes the activation function $\sigma$ regarding logit KD. Table 10 and Figure 4 analyze the impact of the temperature scaling.

**Weaknesses:**

**Limited technical contributions.** The technical novelty of the paper seems quite limited. The paper introduces the Tempered Logit Distillation (TLD) that uses a sigmoid with a temperature parameter and a KL divergence loss. The tempered sigmoid is not novel and was used in published papers like [A]. The theoretical analysis section does not provide theoretical results. It just shows the gradient, and analyzes the impact of the temperature parameter on the learning. As the technical novelty of the paper is quite limited, the theoretical analysis should be improve to include results such as the theoretical convergence guarantees or convergence rate. It could be interesting to build novel analysis tools to better understand the impact of the temperature scaling. In other words, it could be valuable to answer the question: why does the model cannot learn the right scale for the output activations?

**Hard samples.** In the introduction, it is written: "we identify that tempered sigmoid with τ smaller than 1 provides an effect of hard mining by governing the magnitude of penalties according to the sample difficulty, which is shown as the key property to its success." The paper does not justify well this claim. It is not clear why focusing on hard examples is important. The paper should contain an experiment or theoretical result to support this claim. It could also be useful to give a definition of hard examples.

**Things to improve the paper that did not impact the score:**
- Table 1 and 2 show result, but these tables are before the experiment section so it is not possible to understand them well before to read the remaining of the paper. I would suggest to rethink the position of these tables so it is easier for the reader to understand them. For example, showing the an ablation study in the introduction may not be a good idea.
- Some table captions are not self content. For instance, Table 6 and 7 do not indicate the task which can be confusing as COCO is used for three different tasks.

[A] Papernot N, Thakurta A, Song S, Chien S, Erlingsson Ú. Tempered sigmoid activations for deep learning with differential privacy. AAAI 2021.

**Questions:**

The questions are arranged in order of importance, with the first question being the most important.

- The paper or its appendix should explain the derivation of equation 8. The equation in the paper seems quite different from the standard KL gradient. It would be important to explain what is the meaning of $z_i$? $z_i$ is the output logit, but it does not have superscripts, so what is its connection with the teacher and student networks?
- Why focusing on hard examples is important for TLD?
- Why does the model cannot learn the right scale for the output activations and need the temperature scaling?

---

> ### Author Response · Authors · 2024-11-18
> **Response to Reviewer dFkV**
>
> We sincerely thank you for your efforts in reviewing our paper. Due to some comments are coupled or similar, we summarize the following questions and present our answers. Please let us know if your concerns are not totally covered or you have any further issues.
>
> **Q1**. About Equation (8):
> **Q1.1** Explain the derivation of equation (8) in the appendix or main paper. The equation in the paper seems quite different from the standard KL gradient.
> **A1.1**: Thanks for your suggestion! We have added relevant details and descriptions in Appendix C.1
> **Q1.2**: It would be important to explain what is the meaning of $z_{i}$? $z_{i}$ is the output logit, but it does not have superscripts, so what is its connection with the teacher and student networks?
> **A1.2**: Thank you so much for pointing out this unclear part! Yes, $z_{i}$ is the output logit. You are correct. Since we calculate the gradient for students (i.e. the teacher is frozen in KD), it should be$z_{i}^{s}$. Sorry for the confusion raised by this typo, and we have corrected it in our revision.
> ***
> **Q2**. Hard samples: Why focusing on hard examples is important for TLD? and the definition of hard samples.
> **A2**: In KD, as we know, it is hard to give an exact definition to determine which samples/pixels are hard/important for distillation, especially in dense predictive tasks. However, some intuitions are acknowledged and justified by the KD community. We take the object detection task as an example. In detection KD, including our TLD, the informative foreground regions, especially the boundaries between instances or small instances, are generally treated as the hard/important samples [1][2][3][4][5]. In addition, the pixels that exhibit larger output discrepancy between the teacher and student are also deemed as the hard/important areas [5][6][7]. Regulating the student to focus on learning those samples leads to non-trivial improvements [2][3][4][5][6][7]. Similar arguments could also be drawn from the classification task [8][9]. In our paper, as shown in Table 10 and Figure 4, when the distillation process starts assigning large magnitudes of penalties to the useless background, the performance drops immediately. The loss distribution in Figures 4, 5, B.3, 4.C also clearly demonstrates that our TLD with $\tau<1$ certainly provides the effect of hard mining or stressing the areas with larger discrepancy, consequently leading to better performance. Thus, our argument is well justified and aligned with the prior KD related works. We have added a new paragraph in the related work section to highlight the importance of mining hard samples in KD. Last, we hope the reviewer may consider that the primary goal of our work is not to put forth a rule-breaking principle to define the hard/important samples in KD.
>
> [1] Focal Loss for Dense Object Detection, 2017, ICCV
> [2] Focal and Global Knowledge Distillation for Detectors, 2022, CVPR
> [3] Masked Distillation with Receptive Tokens, 2023, ICLR.
> [4] Distilling Object Detectors with Feature Richness, 2021, NeurIPS.
> [5] Bridging Cross-task Protocol Inconsistency for Distillation in Dense Object Detection, 2023, ICCV.
> [6] Knowledge Distillation for Object Detection via Rank Mimicking and Prediction-Guided Feature Imitation, 2022, AAAI
> [7] General Instance Distillation for Object Detection, 2021, CVPR.
> [8] Rethinking Soft Labels for Knowledge Distillation: A Bias-Variance Tradeoff Perspective, 2021, ICLR.
> [9] Less or More from Teacher: Exploring Trilateral Geometry for Knowledge Distillation, 2024, ICLR
> ***
> **Q3**. A new analysis tools to better understanding the impact of the temperature scaling.
> **A3**.: In this manuscript, we study the impact of temperature scaling from the gradient space. As you suggested, we further analyze it by reformulating our distillation loss in Appendix C.2. With the loss reformulation and new visualization (Figure C.4), one may better grasp the impact of the temperature scaling in our method. Specifically, the conclusion is aligned with the one drawn in the manuscript that our method with $\tau<1$ will instruct the student to distill the hard samples.

---

> ### Author Response · Authors · 2024-11-18
> **Response to Reviewer dFkV**
>
> **Q4**: Tempered sigmoid was used in paper [10]
> **A4**. Thanks for bringing up this related paper! After reading [10], we find that our method is substantially different from [10] in the following aspects.
> (1) Different motivation of “sigmoid”: [10] hypothesized that activation functions need to be bounded in private learning and replaced the unbounded ReLU with bounded activation functions (e.g., sigmoid or tanh), where tanh is a variant of sigmoid.
> Paper [10] cares about whether the activation function is bounded or not in private training.
> In contrast, our method is motivated by the different behavior of softmax and sigmoid in multi-label logit KD, and we find that the power of sigmoid has been suppressed for some reasons (i.e., ignoring temperature scaling). Our work focuses on whether the applied activation function is aligned with the attribution of KD and multi-label learning tasks.
>
> (2) Different motivation of “tempered”: Paper [10], inspired by [11], aims to provide robustness to noise during training by temperature scaling. In contrast, our work is partially motivated by the critical role of temperature scaling in logit KD. To the best of our knowledge, its peculiar effect has not been studied in multi-label logit KD so far.
>
> (3) Different technical implementation: Paper [10] replaces ReLU with tempered sigmoid or tanh for both the convolution and fully-connected layers of DNNs. Our method is only involved in the formulation of KD loss, without any modification to the original networks or task loss whatsoever.
> Therefore, our work is totally different from the paper [10], although sharing the concept of “tempered sigmoid”. This embarrassing encounter might not weaken the overall novelty of our work. In our final revision, we will make an effort to collect a new paragraph: “Tempered sigmoid” by looking through papers that also formally meet the tempered sigmoid.
>
> [10] Tempered Sigmoid Activations for Deep Learning with Differential Privacy, 2021, AAAI.
> [11] Robust Bi-Tempered Logistic Loss Based on Bregman Divergences, 2019, NeurIPS.
> ***
> **Q5**.Why does the model cannot learn the right scale for the output activations and need the temperature scaling?
> **A5**: We would like to clarify that the temperature scaling operation is only involved in the distillation loss and the classification task loss still takes the unscaled logit as the input. Keeping the original task loss untouched is a common practice in KD, by which the distillation ability of KD loss can be better reflected.
> ***
> **Q6**: Things to improve the paper, including the table position and table captions
> **A6**: Thanks for your valuable suggestions! We have modified the captions of Tables 6,7,8 and reorganized the paper layout. We agree with you that presenting an ablative study in the introduction section is not a good idea. However, Table 1 is used to support the motivation and empirically answer one key question posed in the introduction. As a compromise, we leave Table 1 in the introduction section.

---

> ### Comment · Reviewer_dFkV · 2024-11-26
>
> Thank you for your detailed responses and for providing the updated paper. I appreciate that the rebuttal addresses most of my concerns. However, it does not fully address my concern regarding the limited technical contribution of the work. Specifically, the use of the sigmoid activation with a temperature parameter, while interesting in the context of the proposed approach, is not novel. This technique has been employed in previous works, albeit for different motivations or in different contexts. For example, [A] utilizes the sigmoid with temperature parameter, and other works such as [B] and [C] also adopt this method. Given this, I have only slightly increased my score.
>
> [A] Papernot N, Thakurta A, Song S, Chien S, Erlingsson Ú. Tempered sigmoid activations for deep learning with differential privacy. AAAI 2021.
>
> [B] Zhai X, Mustafa B, Kolesnikov A, Beyer L. Sigmoid loss for language image pre-training. ICCV 2023
>
> [C] Zhang J, Dai Y, Yu X, Harandi M, Barnes N, Hartley R. Uncertainty-aware deep calibrated salient object detection. 2020

---

> > ### Author Response · Authors · 2024-11-26
> > **Thank you for the further comments!**
> >
> > We sincerely thank you for raising the score and helping us improve the paper so far!  We remain open and responsive to any further discussions until the end of the discussion stage.

---

### Official Review · Reviewer_N27U · 2024-10-29

**Soundness:** 3
**Presentation:** 3
**Contribution:** 2
**Rating:** 6
**Confidence:** 2

**Summary:**

This paper discovers that the vanilla sigmoid does not offer significant practical benefits compared to the tempered softmax in multi-label logit distillation. Furthermore, it identifies that ignoring the temperature parameter is the fundamental bottleneck causing the vanilla sigmoid to yield poor results. Motivated by these findings, the authors introduce the tempered sigmoid and demonstrate its advantages both empirically and theoretically.

**Strengths:**

(1)The writing in this paper is clear and easy to understand.

(2)The quantitative and qualitative results are sufficient.

**Weaknesses:**

The experiments in this paper are primarily conducted on a smaller scale, making it difficult to convince that the method can be generalized to larger datasets, such as those with millions or tens of millions of data points.

**Questions:**

(1) Is the 𝜏 sensitive for different tasks and training sets?

(2) The method proposed in this paper is overly simple, which raises doubts about whether other literature and work has already proposed related or better approaches.

---

> ### Author Response · Authors · 2024-11-18
> **Response to Reviewer N27U**
>
> Thanks for your efforts in reviewing our paper. We address your comments and questions in the following content. If you have any further questions, let us know any time!
>
> **Q1**: Smaller scale datasets and generalization on millions or tens of millions of data points.
>
> **A1**: In terms of methodology, our work is motivated and built upon the intrinsic behavior of activation functions in logit KD, without depending on the size or characteristics of the datasets. In this case, our method may have the potential to be applied to larger datasets. During the rebuttal phase, we honestly do not have enough time/resources to provide empirical results on millions (even tens of millions) level datasets for you. We are sorry for this, but note, that the experiments in our paper are sufficient for validating the effectiveness of the proposed method. In particular, we would like to mention that all the datasets used in this work are the standard benchmark for evaluating KD in the corresponding tasks, as recognized by Reviewer dFKV and mZnK. For example, the COCO is a well-acknowledged and large-scale dataset in multi-label classification, object detection, and instance segmentation.
> ***
> **Q2**: Is the $\tau $ sensitive for different tasks and training sets?
>
> **A2**: As shown in Table 10, our method achieves relatively stable results in a reasonable range of $\tau$ for different tasks, as long as the $\tau$ is smaller than 1. As stated in Appendix A.4 (Training Details), we actually fix the temperature $\tau$ for different training sets and networks.
> ***
> **Q3**: Whether other literature and work have already proposed related or better approaches?
>
> **A3**: As far as we know, this is the first work to investigate the importance of temperature scaling in multi-label logit KD. Please refer to Sec.2 and the contributions part.

---

> > ### Comment · Reviewer_N27U · 2024-11-24
> >
> > I appreciate the author's response. Given the author's feedback, I will retain my original score.

---

> > > ### Author Response · Authors · 2024-11-25
> > > **Further Discussion**
> > >
> > > Dear Reviewer N27U:
> > >
> > > We sincerely thank you for your efforts in reviewing our paper and reading the response. We notice you retain your recommendation, without stating further comments/reasoning. To improve our work in the future, we were wondering and humbly ask 1) if our response does not fully address your concerns, could you generously point the issues out; 2) if you have any further questions, please don't hesitate to reach out. We would like to have detailed discussions and try to address your concerns.
> > >
> > > We appreciate your feedback again!
> > >
> > > Best wishes,
> > >
> > > Authors

---

> > > > ### Author Response · Authors · 2024-12-01
> > > > **Sincerely looking forward to your further feedback!**
> > > >
> > > > Dear Reviewer N27U:
> > > >
> > > > Thank you so much for your time. We kindly remind you that the deadline for the Author/Reviewer Discussion is approaching. We look forward to your further comments and to providing additional clarification if needed. If you prefer to check after this weekend, it is perfectly fine!
> > > >
> > > > Meanwhile, we would like to humbly clarify that the datasets (your major concern) used in our paper are the standard benchmark for evaluating a KD method in academia. We greatly appreciate it if you could generously revisit your evaluation and consider raising your score.
> > > >
> > > > Thank you once again for being with us so far! Have a wonderful weekend!
> > > >
> > > > Best wishes,
> > > >
> > > > Authors, 12/01

---

> > > > > ### Comment · Reviewer_N27U · 2024-12-01
> > > > >
> > > > > Thank you for your response. I believe that experiments on larger datasets would further substantiate the effectiveness of the proposed method, enhance the impact of this paper, and promote advancements in the field. This effort should not be seen as aimed at fulfilling individual requests but rather as benefiting the broader academic community.

---

> > > > > > ### Author Response · Authors · 2024-12-01
> > > > > > **Thank you for following up!**
> > > > > >
> > > > > > Thanks for the suggestion. Beyond the standard experimental setting in prior works, we will consider millions-level datasets in the future work.
> > > > > >
> > > > > > Thank you again for your time.

---

> > > > > > > ### Comment · Reviewer_N27U · 2024-12-03
> > > > > > >
> > > > > > > Based on the author's latest response, my expectations have been met. Consequently, I will increase my rating.

---

### Author Response · Authors · 2024-12-03
**To All Reviewers and Area Chair**

As the rebuttal period comes to the end, we would like to extend our heartfelt gratitude to all of the reviewers for their valuable time and feedback on our manuscript, which have provided effective guidance for improving our paper. We are encouraged that the reviewers recognized the following merits of our work:

**1. Novelty and addressing key limitations**: We clarify a recently discovered perspective is not entirely correct, and, for the first time, explore the peculiar behaviors of temperature scaling in multi-label logit KD. The idea is novel and well-motivated.

**2. Impressive performance and broad applicability**: The proposed method consistently achieves SOTA distillation performance across three multi-label learning tasks (image classification, object detection, and instance segmentation) over three challenging benchmarks (COCO, PASCAL-VOC, and NUS-WIDE).

**3. Theoretical support**: Our work provides theoretical justification, further backing up the proposed method and consummating the behavior of temperature scaling in logit distillation in general.

**4. Reasonable experiments and multiple analyses**: The quantitative and qualitative results are clear and informative, enhancing the reader's understanding of the methodology.

We also sincerely thank the AC in advance for the considerable time and effort that will be devoted to further reviewing our paper.

---

### Note · Authors · 2025-01-26

I have read and agree with the venue's withdrawal policy on behalf of myself and my co-authors.